# NODE CLASSIFICATION BEYOND HOMOPHILY: TOWARDS A GENERAL SOLUTION

## ABSTRACT

Graph neural networks (GNNs) have become core building blocks behind a myriad of graph learning tasks. The vast majority of the existing GNNs are built upon, either implicitly or explicitly, the homophily assumption, which is not always true and could heavily degrade the performance of learning tasks. In response, GNNs tailored for heterophilic graphs have been developed. However, most of the existing works are designed for the specific GNN models to address heterophily, which lacks generality. In this paper, we study the problem from the structure learning perspective and propose a family of general solutions named ALT. It can work hand in hand with most of the existing GNNs to decently handle graphs with either low or high homophily. The core of our method is learning to (1) decompose a given graph into two components, (2) extract complementary graph signals from these two components, and (3) adaptively merge the graph signals for node classification. Moreover, analysis based on graph signal processing shows that our framework can empower a broad range of existing GNNs to have adaptive filter characteristics and further modulate the input graph signals, which is critical for handling complex homophilic/heterophilic patterns. The proposed ALT brings significant and consistent performance improvement in node classification for a wide range of GNNs over a variety of real-world datasets.

## 1 INTRODUCTION

Graph neural networks (GNNs) have demonstrated the great power as building blocks for a variety of graph learning tasks, such as node classification (Kipf & Welling, 2017), graph classification (Xu et al., 2018), link prediction (Zhang & Chen, 2018), clustering (Bianchi et al., 2020), and many more. Most of the existing GNNs follow the homophily assumption, i.e., edges tend to connect nodes with the same labels and similar node features. Such an assumption holds true for networks such as citation networks (Yang et al., 2016; Bojchevski & Günnemann, 2018) where a paper tends to cite related literature. However, in many other cases, the *heterophilic* settings arise. For instance, to form a protein structure, different types of amino acids are more likely to be linked together (Zhu et al., 2020). On such heterophilic networks, the performance of classic GNN models (Klicpera et al., 2018; Veličković et al., 2018; Hamilton et al., 2017) could degrade greatly and might be even worse than an MLP which does not utilize any topology information at all (Zhu et al., 2020).

In response, researchers have analyzed the limitations of the existing GNNs in the presence of node heterophily and further proposed specific models to address it from both the spatial and spectral perspectives. For instance, an important design by H2GCN (Zhu et al., 2020) is that high-order neighbors should be considered during message aggregation. GPRGNN (Chien et al., 2021) also aggregates messages from multi-hop neighbors but it emphasizes that messages can also be negative via a set of learnable aggregation weights. From the spectral perspective, FAGCN (Bo et al., 2021) points out that low-pass filter-based GNNs smooth the node representations between connected nodes, which is not desirable for the heterophilic settings where connected nodes are more likely to have different labels. Hence, FAGCN (Bo et al., 2021) adaptively mixes the low-pass graph filter with the high-pass graph filter via an attention mechanism to tackle this problem. A more detailed review of related work can be found in Section 5.

Despite the theoretic insights and empirical performance gain, most of the existing works focus on the model level, i.e., they aim to propose better GNNs models to handle the heterophilic graphs. In

other words, the success of their methods relies on specific designs of GNN models. In this paper, we take a step further and ask: *how to develop a generic method to benefit a broad range of GNNs for node classification beyond homophily, even if they are not originally tailored for the heterophilic graphs?* To this end, we address this problem from a structure learning (Zhu et al., 2021b) perspective, that is, we optimize the given graph structure to benefit downstream tasks (e.g., node classification). Different from the existing approaches that refine the specific GNNs models, our approach focuses on the data level by optimizing the input graph topology to tackle heterophily.

**Challenges.** In pursuing such a data-centric general solution, here are the key challenges. First (*model diversity*), our goal is to strengthen a broad range of established GNNs so that they can handle graphs with arbitrary homophily. However, the aggregation mechanism and the graph convolution kernels are different between various GNN models. It is unknown how to accommodate diverse GNNs seamlessly. Second (*theoretical foundation*), analyses on the success of some specific GNNs for heterophilic graphs have recently emerged (e.g., from the graph signal processing perspective (Shuman et al., 2013)). However, few works focus on the theoretical foundation of structure learning and its connection to dealing with graphs with low homophily. Our main contributions are listed as follows: (1) We propose a general graph structure learning-based framework named duAL sTructure learning (ALT), which can accommodate a variety of GNN models. Specifically, after removing the activation function from the last layer, *any* GNN can be plugged into our framework and be trained end-to-end with common optimizers. (2) We provide a detailed analysis from the graph signal processing perspective. Our analysis guides the design of ALT and validates its effectiveness theoretically. (3) Experiments show that with the help of ALT, the node classification accuracy of a broad range of existing GNNs is boosted on heterophilic graphs, and meanwhile kept competitive on homophilic graphs.

## 2 PRELIMINARIES

**Notations.** We use bold uppercase letters for matrices (e.g., $\mathbf{A}$), bold lowercase letters for column vectors (e.g., $\mathbf{u}$), lowercase and uppercase letters in regular font for scalars (e.g., $d$, $K$), and calligraphic letters for sets (e.g., $\mathcal{T}$). We use $\mathbf{A}[i, j]$ to represent the entry of matrix $\mathbf{A}$ at the $i$-th row and the $j$-th column, $\mathbf{A}[i, :]$ to represent the $i$-th row of matrix $\mathbf{A}$, and $\mathbf{A}[:, j]$ to represent the $j$-th column of matrix $\mathbf{A}$. Similarly, $\mathbf{u}[i]$ denotes the $i$-th entry of vector $\mathbf{u}$. Superscript $\top$ denotes the transpose of matrices and vectors. $\odot$ denotes the Hadamard product.

An attributed graph can be represented as $\mathcal{G} = \{\mathbf{A}, \mathbf{X}\}$ which is composed of an adjacency matrix $\mathbf{A} \in \mathbb{R}^{n \times n}$ and an attribute matrix $\mathbf{X} \in \mathbb{R}^{n \times d}$, where $n$ is the number of nodes and $d$ is the node feature dimension. In total, nodes can be categorized into a set of classes $C$. The normalized Laplacian matrix is $\tilde{\mathbf{L}} = \mathbf{I} - \mathbf{D}^{-\frac{1}{2}} \mathbf{A} \mathbf{D}^{-\frac{1}{2}}$ where $\mathbf{D}$ is the diagonal degree matrix of $\mathbf{A}$. It can be decomposed as $\tilde{\mathbf{L}} = \mathbf{U} \mathbf{\Lambda} \mathbf{U}^\top$ where $\mathbf{U} \in \mathbb{R}^{n \times n}$ is the eigenvector matrix and $\mathbf{\Lambda} \in \mathbb{R}^{n \times n}$ is the diagonal eigenvalue matrix. In graph signal processing (Shuman et al., 2013), the diagonal entry of $\mathbf{\Lambda}$ represents frequency and $\mathbf{\Lambda}[i, i] = \lambda_i$. Given a signal $\mathbf{x} \in \mathbb{R}^n$, its graph Fourier transform (Shuman et al., 2013) is represented as $\hat{\mathbf{x}} = \mathbf{U} \mathbf{x}$, and its inverse graph Fourier transform is defined as $\mathbf{x} = \mathbf{U}^\top \hat{\mathbf{x}}$. For a diffusion matrix $\mathbf{C} \in \mathbb{R}^{n \times n}$, its frequency response (or profile (Balcilar et al., 2021)) is defined as $\mathbf{\Phi}_{\texttt{fp}} = \texttt{diag}^{-1}(\mathbf{U}^\top \mathbf{C} \mathbf{U})$ where $\texttt{diag}^{-1}(\cdot)$ returns the diagonal entries. This frequency response is also known as the filter and the convolution kernel.

**Semi-supervised Node Classification.** In this paper, we study *semi-supervised node classification* (Yang et al., 2016; Kipf & Welling, 2017) where the graph topology $\mathbf{A}$, all node features $\mathbf{X}$, and a part of node labels are given and our goal is to predict the labels of unlabelled nodes. Numerous works (Kipf & Welling, 2017; Veličković et al., 2018; Klicpera et al., 2018) achieve impressive performance on this problem. However, recent studies show that their successes heavily rely upon the homophily assumption of the given graphs (Zheng et al., 2022; Zhu et al., 2020). In general, homophily describes to what extent edges tend to link nodes with the same labels and similar features. Following previous works (Zhu et al., 2020; Pei et al., 2019), this paper focuses on the node label homophily. There are various homophily metrics and we introduce one of them named edge homophily (Zhu et al., 2020) as: $h(\mathcal{G}) = \frac{\sum_{i,j,\mathbf{A}[i,j]=1} [\![\mathbf{y}[i]=\mathbf{y}[j]]\!]}{\sum_{i,j} \mathbf{A}[i,j]} \in [0, 1]$, where $[\![x]\!] = 1$ if $x$ is true and 0 otherwise. The more homophilic a given graph is, the closer its $h(\mathcal{G})$ is to 1.

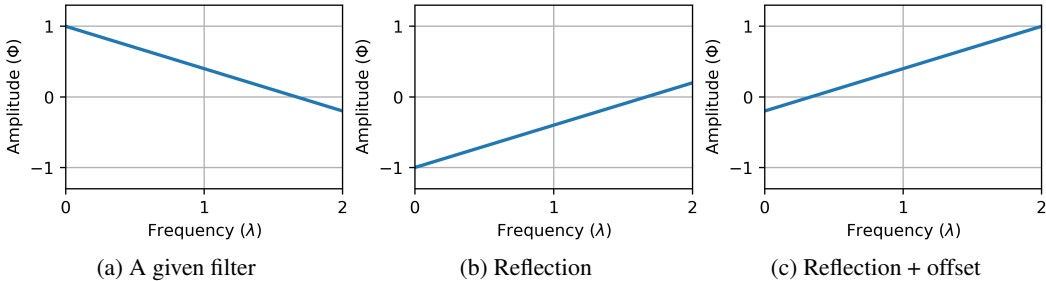

|                    |                    |                         |
| :----------------: | :----------------: | :---------------------: |
| (a) A given filter | (b) Reflection     | (c) Reflection + offset |

Figure 1: The Illustration of obtaining a filter with complementary filter characteristics. Given a filter (a), its reflected frequency response (b) with offset (c) has complementary filter characteristics.

## 3 PROPOSED METHODS

In this section, we first propose a flexible method named ALT-global which empowers any GNN with an **adaptive filter** characteristics. Next, we carefully analyze the expressiveness of ALT-global from the graph signal processing perspective (Shuman et al., 2013). This analysis guides the design of another more advanced method named ALT-local which enhances the spectral expressiveness of any GNN to be a local adaptive filter by **modulating** the input graph signals.

### 3.1 ALT-GLOBAL: A GLOBAL ADAPTIVE METHOD

Intuitively, nodes with different labels should be located as far as possible in the embedding space and nodes with the same labels should be assigned closely. This intuition is aligned well with the utility of many classic GNNs (e.g., GCN (Kipf & Welling, 2017)) on homophilic graphs. That is because, on homophilic graphs, many same-label nodes are connected, whose embeddings will be smoothed by those classic low-pass filter GNNs (Bo et al., 2021; Balcilar et al., 2021). In contrast, the low-pass filter GNNs' performance degrades significantly on heterophilic graphs since the connected nodes' embeddings should not be smoothed. Many efforts (Bo et al., 2021; Chien et al., 2021) point out that a key design to deal with graphs with unknown homophily is to equip GNNs with an adaptive filter.

We aim to propose a data-centric solution such that minimal modification on the given GNNs (e.g., a low-pass filter GNN) is needed. As we do not make any assumption about the model structure of the given GNN, its filter can be either low-pass, high-pass, band-pass, or others. To equip the given GNN with an adaptive filter, our core idea is to *adaptively combine signals from two filters with the complementary filter characteristics*. For example, if a low-pass filter GNN is given, it should be adaptively combined with another high-pass filter. To find such a complementary filter, a two-step modification of the frequency response is needed. Figure 1 shows that we can first *reflect the frequency response curve over the frequency axis* and then *set an appropriate offset to the reflected frequency response*. Guided by this idea, the mathematical details of the proposed ALT-global are as follows,

$$\mathbf{H}_1 = \text{GNN}(w\mathbf{A}, \mathbf{X}, \theta_1), \tag{1a}$$

$$\mathbf{H}_2 = \text{GNN}((1-w)\mathbf{A}, \mathbf{X}, \theta_2), \tag{1b}$$

$$\mathbf{H}_{\text{offset}} = \text{MLP}(\mathbf{X}, \theta_3), \tag{1c}$$

$$\mathbf{Z} = \text{softmax}(\mathbf{H}_1 - \mathbf{H}_2 + \eta\mathbf{H}_{\text{offset}}), \tag{1d}$$

where $\theta_1$ and $\theta_2$ are the parameters of the **backbone dual GNNs** (i.e., GNNs from Eq. 1a and Eq. 1b), $\theta_3$ is the parameter of a multi-layer perceptron (MLP), $\eta \in \mathbb{R}$ and $w \in [0, 1]$ are learnable parameters, and $\mathbf{Z} \in \mathbb{R}^{n \times C}$ is the prediction matrix. Here the softmax is applied row-wise. For models using the normalized adjacency matrix (e.g., $\tilde{\mathbf{A}} = (\mathbf{D} + \mathbf{I})^{-\frac{1}{2}}(\mathbf{A} + \mathbf{I})(\mathbf{D} + \mathbf{I})^{-\frac{1}{2}}$) as the diffusion matrix (e.g., GCN (Kipf & Welling, 2017)), the re-weighting can be set over the normalized adjacency matrix (i.e., $w\tilde{\mathbf{A}}$ and $(1-w)\tilde{\mathbf{A}}$).

We elaborate more on the design of ALT-global. First, all the insights we obtained from Figure 1 are still applicable to the convolution kernel directly. Nonetheless, since our method works in a plug-and-play fashion which does not modify the backbone GNNs, it uses a well-designed aggregation

(i.e., Eq. 1d) to achieve an equivalent effect. Specifically, (1) $\mathbf{H}_1$ is the signals from a backbone GNN with positive re-scaling; (2) $-\mathbf{H}_2$ is the negative signals that correspond to the signals from a reflected filter; (3) $\eta\mathbf{H}_{\texttt{offset}}$ is the offset term which is equivalent to signals from an all-pass filter. Second, the adaptive mixture of the above three sets of graph signals is controlled by the learnable parameters $w$ and $\eta$. Other aggregation functions are also applicable. One of the options is an MLP whose input is the concatenation of $\mathbf{H}_1$, $\mathbf{H}_2$, and $\mathbf{H}_{\texttt{offset}}$. However, it is not used in this paper because (1) it increases the analysis difficulties dramatically and (2) empirically, no performance advantage is observed in the ablation study (Section 4.3). Analysis in the following section shows that ALT-global bears strong flexibility in filter characteristics.

## 3.2 ANALYSIS OF ALT-GLOBAL

For clarity and brevity, in the following analysis, we assume that the backbone GNNs are graph-augmented MLPs (GA-MLPs) as defined below. This is because, first, many GNNs fall into the GA-MLP family if part of the nonlinear functions is removed; and second, GA-MLPs have shown strong empirical performance while enjoying provable expressiveness (Chen et al., 2021).

**Definition 1.** *Graph-Augmented Multi-Layer Perceptron (GA-MLP) (Chen et al., 2021) is a family of GNNs that first conduct feature transformation via an MLP and then diffuse the features. Mathematically they compute node embeddings as* $\mathbf{H} = \mathbf{C} \cdot \texttt{MLP}(\mathbf{X})$ *where* $\mathbf{C}$ *is the diffusion matrix.*

The (full) frequency profile (Balcilar et al., 2021) is closely related to the filter characteristics of GNNs and it is introduced as follows.

**Definition 2.** *Frequency profile (Balcilar et al., 2021) is defined as* $\mathbf{\Phi}_{fp} = \texttt{diag}^{-1}(\mathbf{U}^\top\mathbf{CU})$ *where* $\texttt{diag}^{-1}(\cdot)$ *returns the diagonal entries if* $\mathbf{U}^\top\mathbf{CU}$ *is a diagonal matrix. In case* $\mathbf{U}^\top\mathbf{CU}$ *is not a diagonal matrix,* ***full frequency profile*** *(Balcilar et al., 2021) is defined as* $\mathbf{\Phi} = \mathbf{U}^\top\mathbf{CU}$.

It is well-known that the frequency profile of a diffusion matrix (if diagonal) is a filter/convolution kernel for the input graph signal. Next, we show that ALT is indeed equipped with an adaptive filter.

**Lemma 1.** *The filter characteristic of the proposed* ALT-*global (Eq. 1d) is adaptive regardless of the frequency filtering functionality of the backbone GNNs (Eq. 1a and Eq. 1b).*

*Proof.* For analysis convenience, we assume (1) the learnable weight $w$ is multiplied with the diffusion matrix, and (2) the backbone GNNs are GA-MLPs whose MLP modules (from Eq. 1a and Eq. 1b) share common parameters with the offset MLP (from Eq. 1c). We start from the case where backbone GNNs are fixed low-pass filters. Without loss of generality, their corresponding full frequency profiles can be presented as $\mathbf{\Phi} = \mathbf{I} - \xi(\mathbf{\Lambda})$ where $\xi$ is a monotonically increasing function. Then, in this case, the diffusion matrices from two GNNs are re-weighted as $w\mathbf{C}$ and $(1-w)\mathbf{C}$ respectively. Considering the offset MLP as a special GA-MLP whose diffusion matrix is $\mathbf{I}$, the combined graph signals are $w\mathbf{C} \cdot \texttt{MLP}(\mathbf{X}) - (1-w)\mathbf{C} \cdot \texttt{MLP}(\mathbf{X}) + \eta\mathbf{I} \cdot \texttt{MLP}(\mathbf{X}) = \tilde{\mathbf{C}} \cdot \texttt{MLP}(\mathbf{X})$ where the combined diffusion matrix is $\tilde{\mathbf{C}} = w\mathbf{C} - (1-w)\mathbf{C} + \eta\mathbf{I}$. Hence the diagonal entry of the corresponding full frequency profile is

$$\mathbf{\Phi}[i,i] = \Phi(\lambda_i) = (2w-1)(1-\xi(\lambda_i)) + \eta.$$

When $w > 0.5$, i.e., $2w - 1 > 0$, $\Phi(\lambda_i)$ is a monotonically decreasing function. The proposed method is a low-pass filter when $\eta > 0$. Similarly, it is a high-pass filter when $w$ is close to $0$ and $\eta > 1$. The above conditions are sufficient and in fact, there are many other combinations of $w$ and $\eta$ which can produce low-pass/high-pass filters. Similar results can be obtained when the backbone GNNs are fixed high-pass filters and we omit that part for brevity. $\square$

*Remarks.* The filter characteristics of the ALT-global can also be interpreted from the Graph Diffusion Equation (GDE) (Newman, 2018) perspective and we provide the GDE-related analysis in Appendix.

## 3.3 GLOBAL FILTERS VS. LOCAL FILTERS

We have shown that ALT-global is equipped with adaptive filter characteristics. However, ALT-global fundamentally applies a global filter to every node, which could lead to suboptimal performance. Recent studies (Zhu et al., 2021a; Wang et al., 2022a) reveal that heterophilic connection patterns

differ between different nodes. Take gender classification on a dating network as an example. While node pairs are often of different labels (i.e., genders), homosexuality also exists between some node pairs. Therefore, simply applying a global low-pass or high-pass filter over all the nodes can degrade the overall classification performance.

Next, we will study how to generalize our proposed ALT-global to a local (i.e., node-specific) and adaptive filter. Before that, let us take a closer look at the full frequency profile (Balcilar et al., 2021): $\mathbf{\Phi} = \mathbf{U}^\top \mathbf{C} \mathbf{U}$. In the following proposition, we point out that $\mathbf{\Phi}$ can describe both the filter and modulator characteristics of a given diffusion matrix $\mathbf{C}$.

**Proposition 1.** *The diagonal entries of the full frequency profile $\mathbf{\Phi}$ of the diffusion matrix serve as the **filter** and the non-zero off-diagonal entries are the **frequency modulator**.*

*Proof.* The diffusion of the input graph signal $\mathbf{X}_{\text{in}} = \texttt{MLP}(\mathbf{X})$ can be represented as $\mathbf{C}\mathbf{X}_{\text{in}} = \mathbf{U}\mathbf{\Phi}\mathbf{U}^\top\mathbf{X}_{\text{in}} = \mathbf{U}(\mathbf{\Phi}\hat{\mathbf{X}}_{\text{in}})$, where $\hat{\mathbf{X}}_{\text{in}}$ is the input graph signal in spectral domain. According to the definitions of graph signal processing (Shuman et al., 2013), $(\mathbf{\Phi}\hat{\mathbf{X}}_{\text{in}})[i:]$ represents the amplitude of output graph signal whose frequency is $\lambda_i$. We further expand the computation and obtain

$$(\mathbf{\Phi}\hat{\mathbf{X}}_{\text{in}})[i:] = \sum_j \mathbf{\Phi}[i,j] \cdot \mathbf{X}_{in}[j,:].$$

In the summation, if $i = j$, it represents the filter/convolution kernel which has been adopted by many spectral GNNs (Balcilar et al., 2021). If $i \neq j$ (i.e., if non-zero off-diagonal entries of $\mathbf{\Phi}$ exist), it shows that the $\lambda_i$-component of the output graph signal is merged with scaled (by $\mathbf{\Phi}[i,j]$) $\lambda_j$-component of the input graph signal which is essentially the modulation (Shuman et al., 2013). $\square$

Based on the above property of the full frequency profile $\mathbf{\Phi}$, the following proposition points out the key design for local filter characteristics.

**Proposition 2.** *Modulation of the input graph signal (i.e., **non-zero off-diagonal entries** in the full frequency profile) is necessary for **local** filters.*

*Proof.* We follow the terminology used in the proof of Proposition 1. If the full frequency profile $\mathbf{\Phi}$ only contains non-zero diagonal entries, we can obtain

$$(\mathbf{\Phi}\hat{\mathbf{X}}_{\text{in}})[i,:] = (\texttt{diag}^{-1}(\mathbf{\Phi}))^\top \odot \hat{\mathbf{X}}_{\text{in}}[i,:], \tag{2}$$

where $\texttt{diag}^{-1}$ extracts the diagonal entries into a vector from the input square matrix. Hence, if we define the scaling of the $\lambda_i$-frequency signal over node $p$ after and before the operator $\mathbf{\Phi}$ as $\texttt{SCALING}(i,p,\mathbf{\Phi}) = \frac{(\mathbf{\Phi}\hat{\mathbf{X}}_{\text{in}})[i,p]}{\hat{\mathbf{X}}_{\text{in}}[i,p]}$, based on Eq. 2 we obtain

$$\forall i, p, q, \quad \texttt{SCALING}(i,p,\mathbf{\Phi}) = \texttt{SCALING}(i,q,\mathbf{\Phi})$$

i.e., for any specific frequency (e.g., $\lambda_i$), its scaling over any two nodes ($p$ and $q$) are equal. In other words, the filter $\mathbf{\Phi}$ works globally over every node. If we expect the filter $\mathbf{\Phi}$ to not work globally, i.e.

$$\exists i, p, q, \quad \texttt{SCALING}(i,p,\mathbf{\Phi}) \neq \texttt{SCALING}(i,q,\mathbf{\Phi}).$$

The above inequality is equivalent to

$$\frac{\sum_{k,k\neq i}\mathbf{\Phi}[i,k] \cdot \hat{\mathbf{X}}_{\text{in}}[k,p]}{\hat{\mathbf{X}}_{\text{in}}[i,p]} \neq \frac{\sum_{k,k\neq i}\mathbf{\Phi}[i,k] \cdot \hat{\mathbf{X}}_{\text{in}}[k,q]}{\hat{\mathbf{X}}_{\text{in}}[i,q]}.$$

Assume that $\forall k$, if $k \neq i$, $\mathbf{\Phi}[i,k] = 0$, and then the left-hand side is equal to the right-hand side which leads to a contradiction. Hence, non-zero off-diagonal entries of the full frequency profile $\mathbf{\Phi}$ must exist if we expect the filter to not work globally. Notice that the above definition of scaling (e.g., $\frac{(\mathbf{\Phi}\hat{\mathbf{X}}_{\text{in}})[i,p]}{\hat{\mathbf{X}}_{\text{in}}[i,p]}$) is not fully aligned with the classic graph filtering (Shuman et al., 2013) but a combination of filtering and modulation as we mentioned in Proposition 1. $\square$

Next, we present a family of GA-MLPs whose spectral expressiveness is limited to a global filter.

**Proposition 3.** *A family of GA-MLPs are global filters if their full frequency profiles are in the form of $\mathbf{C} = \sum_k a_k \tilde{\mathbf{A}}^k + b\mathbf{I}$ which only contains non-zero diagonal entries.*

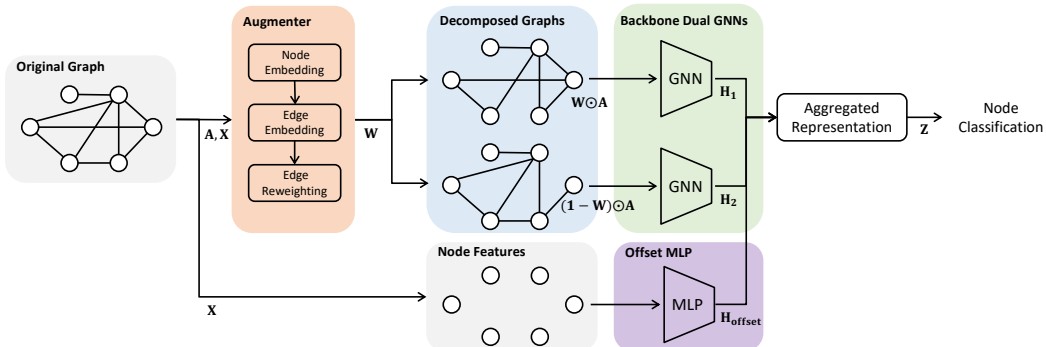

Figure 2: The proposed ALT-local.

We prove Proposition 3 in the Appendix. A wide range of GA-MLPs (e.g., SGC (Wu et al., 2019), APPNP (Klicpera et al., 2018)) follow the above form and therefore cannot modulate graph signal. Unfortunately, even when they are equipped with our proposed ALT-global, they are still *global* filters because ALT-global assigns the same weight to every edge (i.e., $w\tilde{\mathbf{A}}$ and $(1-w)\tilde{\mathbf{A}}$).

## 3.4 ALT-LOCAL: A LOCAL ADAPTIVE METHOD

In this subsection, we propose a more flexible method based on ALT-global. Our goal is to empower the backbone GNNs with local adaptive signal filtering capabilities, which is an essential property for capturing complex heterophilic connection patterns. (Zhu et al., 2021a; Wang et al., 2022a). According to Proposition 3, we know that if all the edges are assigned with the same weight (e.g., $w\tilde{\mathbf{A}}$) the corresponding full frequency profile will only contain diagonal non-zero entries. Lemma 2 provides a clue on how to bring non-zero off-diagonal entries in full frequency profiles.

**Lemma 2.** *By re-weighting the edge weights non-uniformly (i.e., if re-weighting by $\mathbf{W} \odot \tilde{\mathbf{A}}$, $\exists i,j,k,l, \mathbf{W}[i,j] \neq \mathbf{W}[k,l]$), the off-diagonal entries of $\mathbf{\Phi}$ can be non-zero.*

We prove Lemma 2 in the appendix. Guided by Lemma 2 we modify ALT-global as follows so that the edge weights are different:

$$\mathbf{H}_1 = \texttt{GNN}(\mathbf{W} \odot \mathbf{A}, \mathbf{X}, \theta_1), \tag{3a}$$

$$\mathbf{H}_2 = \texttt{GNN}((\mathbf{1} - \mathbf{W}) \odot \mathbf{A}, \mathbf{X}, \theta_2), \tag{3b}$$

$$\mathbf{H}_{\texttt{offset}} = \texttt{MLP}(\mathbf{X}, \theta_3), \tag{3c}$$

$$\mathbf{Z} = \texttt{softmax}(\mathbf{H}_1 - \mathbf{H}_2 + \eta\mathbf{H}_{\texttt{offset}}), \tag{3d}$$

One option is to set $\mathbf{W}$ as a learnable parameter which is prune to overfitting as the number of parameters is equal to the number of edges. Therefore, we parameterize the edge weight $\mathbf{W}$ by an edge augmenter as follows,

$$\mathbf{H} = \texttt{GNN}_{\texttt{aug}}(\mathbf{A}, \mathbf{X}, \phi_1), \tag{4a}$$

$$\mathbf{W}[i,j] = w_{ij} = \texttt{sigmoid}(\texttt{MLP}(\mathbf{H}[i,:]||\mathbf{H}[j,:], \phi_2)) \tag{4b}$$

where $\phi_1$ and $\phi_2$ are the parameters of the augmenter GNN and a multi-layer perceptron (MLP) respectively. Here we first obtain the node embedding matrix via the augmenter GNN (i.e., $\texttt{GNN}_{\texttt{aug}}$) in Eq. 4a. Then we concatenate node embeddings into edge embeddings (i.e., $\mathbf{H}[i,:]||\mathbf{H}[j,:]$). The edge weight (i.e., $w_{ij}$) is computed via an MLP with $\texttt{sigmoid}$ activation. Naturally, the node embeddings from the augmenter GNN (Eq. 4a) should be as discriminative as possible so that the edge importance can be better measured. Thus, we use a two-layer high-pass filter GNN as the $\texttt{GNN}_{\texttt{aug}}$ whose mathematical formulation is as follows,

$$\texttt{GNN}_{\texttt{aug}}(\mathbf{A}, \mathbf{X}, \phi_1) = \tilde{\mathbf{A}}^2_{\texttt{high}}\texttt{MLP}(\mathbf{X}, \phi_1), \tag{5a}$$

$$\tilde{\mathbf{A}}_{\texttt{high}} = \epsilon\mathbf{I} - \mathbf{D}^{-\frac{1}{2}}\mathbf{A}\mathbf{D}^{-\frac{1}{2}}, \tag{5b}$$

where $\epsilon$ is a scaling hyper-parameter to adjust the amplitude of the high-pass filter. We name the above model (i.e., Eqs.3a- 5b) as ALT-local which is summarized in Figure 2.

*Remarks.* Our method is partly inspired by FAGCN (Bo et al., 2021) and we claim the uniqueness and advantages of our work compared with FAGCN as follows. From the method perspective, FAGCN explicitly mixes high-frequency and low-frequency signals. ALT generalizes this idea to the 'mixture of complementary filters'; thus, even though the backbone GNN's convolution kernel is unknown, ALT can still boost its performance decently, which provides great generality. For the theoretical contribution, Bo et al. (2021) analyze the spatial effects of signals with different frequencies. Our analysis takes a solid step forward to reveal the connections between the full frequency profile, graph signal modulation, and local adaptive filters.

### 3.5 TRAINING PROCEDURE

To train our models, we formulate the following bi-level optimization problem.

$$\phi^* = \arg\min_{\phi} \mathcal{L}_{\texttt{upper}}(g(\mathcal{G}, \phi), \theta^*, \mathcal{Y}_{\texttt{valid}}) \quad \text{s.t. } \theta^* = \arg\min_{\theta} \mathcal{L}_{\texttt{lower}}(g(\mathcal{G}, \phi), \theta, \mathcal{Y}_{\texttt{train}}), \quad (6)$$

where the augmenter is denoted as $g(\cdot)$ whose parameter is $\phi$ and the dual backbone GNNs are parameterized as $\theta$ for brevity. Specifically, for the ALT-global, $\theta = \{\theta_1, \theta_2, \theta_3\}$ and $\phi = w$ are from Eq.1a, Eq.1b, and Eq.1c. For ALT-local, $\theta = \{\theta_1, \theta_2, \theta_3\}$ is from Eq. 3a, 3b, and Eq. 3c; $\phi = \{\phi_1, \phi_2\}$ is from Eq. 4a and 4b. Both $\mathcal{L}_{upper}$ and $\mathcal{L}_{lower}$ are cross-entropy loss between the classification results (Eq. 1d for ALT-global and Eq. 3d for ALT-local) and the labelled nodes. The difference lies in that, for the lower-level objective we compute the loss over the training nodes but for the upper-level one we compute the loss over the validation nodes. To solve such a bilevel optimization problem, we resort to the classic first-order approximation (Nichol et al., 2018) to compute the hyper-gradient $\nabla_{\phi} \mathcal{L}_{upper}$ and any gradient descent-based methods can then be used.

If all the feature dimensions of different layers (including the input layers) from different backbone GNNs and MLPs are denoted as $d$ and all the models (GNNs and MLPs) contain 2 feature transformation matrices, the number of trainable parameters of ALT-local is composed of three parts: (1) $\texttt{GNN}_{\texttt{aug}}$ $(2d^2)$, (2) MLP from Eq. 4b $(2d^2 + d)$, (3) $\texttt{GNN}_1$, $\texttt{GNN}_2$, and offset MLP $(3d^2 + 3dc)$ where $c$ is the number of classes. In practice, the parameter number is much smaller than the estimated number. For example for datasets whose $d > 500$, empirically, setting the hidden dimension as 32 is enough. However, compared with vanilla backbone GNNs (e.g., a simple GCN (Kipf & Welling, 2017)), ALT-local inevitably contains more parameters as ALT-local is composed of 3 GNNs and 2 MLPs in total. Even for ALT-global, it is still composed of 2 GNNs and 1 MLP. Hence, the increased number of parameters is a potential limitation of ALT-local and ALT-global. Besides, our theoretical analysis relies on the assumption that the backbone GNNs are GA-MLPs. Generalizing our theoretical results to a broader range of GNNs is our future work.

## 4 EXPERIMENTS

### 4.1 EXPERIMENT SETTINGS

**Datasets.** We use 16 datasets, including Cora (Yang et al., 2016), Citeseer (Yang et al., 2016), Pubmed (Yang et al., 2016), DBLP (Bojchevski & Günnemann, 2018), Computers (Shchur et al., 2018), Photos (Shchur et al., 2018), CS (Shchur et al., 2018), Physics (Shchur et al., 2018), Cornell (Pei et al., 2019), Texas (Pei et al., 2019), Wisconsin (Pei et al., 2019), Chameleon (Rozemberczki et al., 2021), Squirrel (Rozemberczki et al., 2021), Film (Pei et al., 2019), Cornell5 (Lim et al., 2021), and Penn94 (Lim et al., 2021). For Cora, Citeseer, and Pubmed, we follow the dataset split from (Kipf & Welling, 2017). We randomly split the other datasets into $20/20/60\%$ for training, validation, and test. Detailed statistics of the datasets are presented in the Appendix - Dataset Statistics.

**Baseline Methods and Metric.** We use 6 baseline methods including 3 classic GNNs: GCN (Kipf & Welling, 2017), SGC (Wu et al., 2019), and APPNP (Klicpera et al., 2018), and 3 adaptive GNNs: GPRGNN (Chien et al., 2021), FAGCN (Bo et al., 2021), and H2GCN (Zhu et al., 2020) which use specific designs to tackle graphs with low homophily. Thanks to the flexibility of our method, we equip the above baseline methods with our proposed ALT to validate the effectiveness. As ALT-local is more powerful than ALT-global, we mainly show the performance comparison with ALT-local (short as ALT). The comparison between ALT-local and ALT-global will be presented in the ablation study. We use the accuracy (ACC) as the metric and report the average accuracy with the standard deviation in 10 runs.

Table 1: Performance comparison (mean±std accuracy) on heterophilic graphs. The last column indicates the average performance boosting for a specific backbone GNN over all the datasets.

| Backbone | ALT? | Chameleon | Squirrel | Texas | Wisconsin | Cornell | Film | Cornell5 | Penn94 | Avg. Δ |
|---|---|---|---|---|---|---|---|---|---|---|
| GCN | No | 58.4±0.4 | 35.4±0.6 | 57.6±3.5 | 44.4±1.6 | 55.9±0.6 | 28.1±0.3 | 72.8±0.2 | 75.1±0.4 | +10.5 |
|  | Yes | 61.6±0.9 | 41.9±0.4 | 70.3±1.1 | 79.0±0.7 | 73.9±1.4 | 34.8±0.3 | 73.5±0.3 | 76.9±0.8 |  |
| SGC | No | 58.4±0.6 | 36.8±0.4 | 58.6±1.9 | 46.5±1.8 | 57.0±0.4 | 27.3±0.1 | 73.5±0.3 | 75.7±0.2 | +10.6 |
|  | Yes | 61.9±0.8 | 42.2±1.0 | 70.0±0.4 | 82.0±0.9 | 78.4±0.6 | 32.9±0.2 | 74.4±0.4 | 77.4±0.4 |  |
| APPNP | No | 43.0±0.9 | 24.2±0.4 | 59.5±1.1 | 45.7±2.0 | 56.3±1.4 | 28.7±0.3 | 72.2±0.2 | 74.0±0.1 | +11.5 |
|  | Yes | 55.0±0.7 | 33.5±1.7 | 72.1±1.7 | 76.3±3.3 | 75.0±1.6 | 33.6±0.5 | 74.1±0.1 | 75.7±0.5 |  |
| GPRGNN | No | 59.2±0.5 | 38.4±0.8 | 69.1±1.0 | 72.4±1.6 | 69.6±2.5 | 31.3±1.1 | 74.1±0.4 | 78.6±0.4 | +2.5 |
|  | Yes | 59.4±1.2 | 38.2±0.9 | 76.3±1.5 | 79.5±0.7 | 70.9±2.9 | 32.1±0.8 | 75.9±0.3 | 80.4±0.2 |  |
| FAGCN | No | 54.3±1.9 | 32.5±1.4 | 61.5±1.3 | 56.6±5.2 | 66.0±1.7 | 33.8±0.7 | 69.6±0.8 | 73.5±0.4 | +4.1 |
|  | Yes | 57.3±1.0 | 35.6±1.8 | 66.9±3.9 | 69.0±1.8 | 67.9±4.9 | 36.1±0.3 | 71.7±0.9 | 75.8±0.8 |  |
| H2GCN | No | 49.9±0.4 | 29.8±0.8 | 65.8±2.1 | 69.5±2.1 | 63.7±0.4 | 34.5±0.3 | 70.8±0.4 | 73.9±0.3 | +4.2 |
|  | Yes | 54.0±0.3 | 35.3±0.9 | 72.4±2.8 | 77.7±0.3 | 68.5±3.5 | 34.4±0.4 | 73.2±0.1 | 76.2±0.4 |  |

Table 2: Performance comparison (mean±std accuracy (%)) on homophilic graphs. The last column indicates the average performance boosting for a specific backbone GNN over all the datasets.

| Backbone | ALT? | Cora | Citeseer | Pubmed | DBLP | Computers | Photos | CS | Physics | Avg. Δ |
|---|---|---|---|---|---|---|---|---|---|---|
| GCN | No | 81.1±0.3 | 71.2±0.7 | 79.0±0.4 | 83.7±0.1 | 66.2±1.0 | 84.1±0.5 | 88.2±0.2 | 95.3±0.1 | +2.5 |
|  | Yes | 80.9±0.5 | 71.5±0.2 | 79.2±0.3 | 83.4±0.1 | 77.8±0.4 | 88.4±0.1 | 92.0±0.2 | 95.6±0.1 |  |
| SGC | No | 80.8±0.1 | 71.0±0.2 | 79.5±0.5 | 83.8±0.0 | 69.1±0.4 | 86.2±0.4 | 89.7±0.1 | 95.3±0.0 | +2.1 |
|  | Yes | 80.6±0.5 | 71.3±0.1 | 79.6±0.4 | 83.1±0.2 | 79.9±0.3 | 88.6±1.5 | 92.8±0.1 | 95.9±0.0 |  |
| APPNP | No | 82.1±0.1 | 71.8±0.1 | 79.8±0.5 | 83.8±0.2 | 66.7±1.1 | 83.4±1.2 | 87.8±0.1 | 94.9±0.0 | +2.7 |
|  | Yes | 82.4±0.4 | 71.7±0.2 | 79.5±0.8 | 84.4±0.1 | 77.6±0.8 | 88.3±0.7 | 92.4±0.4 | 95.5±0.1 |  |
| GPRGNN | No | 78.6±1.5 | 68.9±0.9 | 77.6±0.9 | 84.4±0.2 | 85.0±0.5 | 92.4±0.2 | 92.3±0.1 | 95.5±0.4 | +0.6 |
|  | Yes | 80.9±0.3 | 68.8±0.2 | 78.2±0.4 | 84.4±0.3 | 85.9±1.5 | 92.6±0.3 | 93.2±0.2 | 95.7±0.1 |  |
| FAGCN | No | 79.0±0.6 | 72.1±0.5 | 78.0±1.1 | 81.1±1.1 | 74.8±3.4 | 91.2±0.3 | 93.0±1.4 | 95.7±0.3 | +0.5 |
|  | Yes | 79.0±0.4 | 71.9±0.5 | 77.9±0.5 | 82.5±0.7 | 76.1±3.9 | 91.9±0.7 | 93.6±0.2 | 96.0±0.1 |  |
| H2GCN | No | 78.9±0.6 | 70.3±1.0 | 78.2±1.0 | 82.4±0.0 | 75.8±0.3 | 89.7±0.2 | 92.5±0.2 | 96.2±0.1 | +0.8 |
|  | Yes | 79.3±0.7 | 70.5±1.3 | 77.9±1.0 | 82.2±0.2 | 80.4±0.8 | 90.3±1.0 | 93.5±0.2 | 96.3±0.1 |  |

## 4.2 MAIN RESULTS

We present the performance comparison on heterophilic graphs in Table 1. First, on the heterophilic graphs, in general, our method ALT can significantly improve the performance of most of the existing GNNs, especially for methods originally not designed for the heterophilic graphs (e.g., GCN, SGC, and APPNP). On average, over $10\%$ improvement is obtained among the heterophilic graphs. Second, over the heterophilic graphs, for adaptive GNNs (e.g., GPRGNN, FAGCN, and H2GCN), their performance improvement is not as significant as low-pass filter GNNs. This is expected since these methods have already dealt with heterophily to some extent. Nonetheless, we still gain $2-4\%$ performance improvements averaged over all 8 heterophilic datasets.

The performance comparison on homophilic graphs is presented in Table 2. We test 48 graph-GNN combinations, out of which, 35 cases show improvements. It is worth noting that even though GCN, SGC, and APPNP are designed mainly for homophilic graphs, the proposed ALT is still able to significantly boost their performance on Computers by nearly $10\%$. Moreover, for each backbone GNN, the average gain of applying the proposed ALT over all 8 homophilic graphs is always positive. Most of the remaining cases bear very minor performance losses (12 out of 13 are below $0.5\%$). Thus, we conclude that ALT can retain or even boost the performance of given backbone GNNs on homophilic graphs.

## 4.3 ABLATION STUDY AND HYPERPARAMETER STUDY

In this section, we present a systematic ablation study on datasets: Chameleon (Rozemberczki et al., 2021), Squirrel (Rozemberczki et al., 2021), Film (Pei et al., 2019), Computers (Shchur et al., 2018), Photos (Shchur et al., 2018), and CS (Shchur et al., 2018). Specifically, we have the following ablated versions: (1) ALT-local, (2) ALT-local with a low-pass filter augmenter (i.e., change Eq.5b as a two-layer SGC) which is named as ALT-local-low, (3) ALT-local-concat whose aggregation step (Eq. 3d)

Table 3: Results of ablation study (Backbone GNN: GCN).

| Backbone | Version | Chameleon | Squirrel | Film | Computers | Photos | CS |
|---|---|---|---|---|---|---|---|
| GCN | None | 58.4±0.4 | 35.4±0.6 | 28.1±0.3 | 66.2±1.0 | 84.1±0.5 | 88.2±0.2 |
| | Global | 58.5±1.2 | 36.5±0.5 | 30.1±0.2 | 67.5±1.1 | 85.9±0.7 | 89.5±0.1 |
| | Local-low | 60.4±0.6 | 39.8±0.9 | 31.6±0.3 | 73.4±1.5 | 86.2±0.6 | 90.4±0.3 |
| | Local-concat | 43.3±1.8 | 29.2±0.6 | 34.7±1.0 | 72.1±4.0 | 85.1±4.1 | 87.2±0.8 |
| | Local | **61.6±0.9** | **41.9±0.4** | **34.8±0.3** | **77.8±0.4** | **88.4±0.1** | **92.0±0.2** |

is instantiated by 'concatenation' followed by an MLP (4) ALT-global, and (5) vanilla backbone GNNs without our methods (named as None). Results with GCN as the backbone are presented in Table 3 and results with SGC and APPNP as the backbones are presented in the Appendix - Additional Experimental Results. From the above results we conclude that the ALT-local has consistent advantages over all ablated versions. In addition, we provide a hyperparameter sensitivity study in the Appendix - Additional Experimental Results.

## 5    RELATED WORK

**Graph Structure Learning.** Graph structure learning aims to modify the given graph structure to improve the performance of downstream tasks. For instance, to boost message propagation, inserting virtual nodes is an effective approach (Gilmer et al., 2017; Li et al., 2017). For topology denoising, dropping some existing edges can improve the model robustness (Wu et al., 2020; Luo et al., 2021) and eliminate redundant information from the input (Yu et al., 2020). Another line of research views the given graph as the optimization variable and updates them according to the performance of downstream node classifiers (e.g., LDS (Franceschi et al., 2019) and Gasoline (Xu et al., 2022)). Other works which formulate the given graph as a random variable and infer its optimal parameters include Bayesian GCNN (Zhang et al., 2019), GEN (Wang et al., 2021), and many more. Recently, Zhu et al. (Zhu et al., 2021b) provide a comprehensive survey on this topic.

**Graph Learning on Heterophilic Graphs.** Heterophilic graphs are also known as disassortative graphs. Many message-passing based GNNs suffer from the performance degradation on the heterophilic graphs and several approaches have been developed for that. For example, Geom-GCN (Pei et al., 2019) and H2GCN (Zhu et al., 2020) expand the message-passing mechanism beyond the first-order neighbors. GPRGNN (Chien et al., 2021) and BernNet (He et al., 2021) set the weights for different propagation results as learnable parameters to work as an adaptive graph filter. FAGCN (Bo et al., 2021), GBK (Du et al., 2022), and ACM-GNN (Luan et al., 2021) explicitly mix two convolution kernels through attention-based mechanisms. Based on the above work, DMP (Yang et al., 2021) studies this problem in a finer granularity where it introduces a feature specific message-passing mechanism. Yan et al. (2021) reveal the connections between oversmoothing and network heterophily. Other works which modify the propagation step of GNNs for the heterophilic graphs include CPGNN (Zhu et al., 2021a), HOG-GCN (Wang et al., 2022b), and GloGNN (Li et al., 2022). Interestingly, Luan et al. (Luan et al., 2021) and Ma et al. (Ma et al., 2021) both report that there are some cases where high heterophily will not hurt the performance of low-pass filter GNN which reveals further unexplored space for this problem. Zheng et al. (Zheng et al., 2022) recently present a survey on this topic. The only structure learning-based solution on addressing graph heterophily, as far as authors' knowledge, is WRGAT (Suresh et al., 2021) which improves the graph homophily by a heuristic method. As a comparison, our proposed framework is more flexible and theoretically solid.

## 6    CONCLUSION

In this paper, we propose a general framework ALT for the semi-supervised node classification problem on graphs beyond homophily. Our method introduces a novel structure learning-based augmenter to decompose the given graph. After that, a dual GNN module can be instantiated as most of the existing GNNs on the decomposed graphs. Systematic theoretical analysis shows that our proposed method can adaptively filter and modulate the graph signals which is critical to address complex heterophilic connection patterns. Comprehensive empirical evaluation and ablation study demonstrate that the proposed ALT obtains significant performance improvement for a wide range of GNN models, on a variety of graph datasets with arbitrary homophily.

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

## A    REPRODUCIBILITY

**Hardware**    We implement ALT  in pytorch[1] and pytorch-geometric [2] using one NVIDIA Tesla V100 SXM2-32GB.

**Dataset statistics**    The detailed statistics of datasets are presented in Table 4 and Table 5.

Table 4: Dataset statistics of heterophilic graphs.

|  | **Chameleon** | **Squirrel** | **Texas** | **Wisconsin** | **Cornell** | **Film** | **Cornell5** | **Penn94** |
|---|---|---|---|---|---|---|---|---|
| # Nodes | 2,277 | 5,201 | 183 | 251 | 183 | 7,600 | 18,660 | 41,554 |
| # Edges | 62,792 | 396,846 | 325 | 515 | 298 | 30,019 | 1,581,554 | 2,724,458 |
| # Features | 2,325 | 2,089 | 1,703 | 1,703 | 1,703 | 932 | 4,735 | 4,814 |
| # Classes | 5 | 5 | 5 | 5 | 5 | 5 | 2 | 2 |
| $h(\mathcal{G})$ | 0.231 | 0.222 | 0.108 | 0.196 | 0.305 | 0.219 | 0.479 | 0.470 |

Table 5: Dataset Statistics of homophilic graphs.

|  | **Cora** | **Citeseer** | **Pubmed** | **DBLP** | **Computers** | **Photos** | **CS** | **Physics** |
|---|---|---|---|---|---|---|---|---|
| # Nodes | 2,708 | 3,327 | 19,717 | 17,716 | 13,752 | 7,650 | 18,333 | 34,493 |
| # Edges | 10,556 | 9,104 | 88,648 | 105,734 | 491,722 | 238,162 | 163,788 | 495,924 |
| # Features | 1,433 | 3,703 | 500 | 1,639 | 767 | 745 | 6,805 | 8,415 |
| # Classes | 7 | 6 | 3 | 4 | 10 | 8 | 15 | 5 |
| $h(\mathcal{G})$ | 0.810 | 0.736 | 0.802 | 0.828 | 0.777 | 0.827 | 0.808 | 0.931 |

**Detailed Experimental Settings**    We obtain all the datasets from pytorch-geometric [3] which are public. We follow the given dataset split for Cora, Citeseer, and Pubmed. For the remaining datasets, we randomly split them into 20/20/60% as training, validation, and test set. Notice that here we do not follow the dataset split from the paper of GPRGNN (Chien et al., 2021) as they manually assign the same number of training samples to each class and our dataset split is more practical. For all the GNNs (including the augmenter and backbone GNNs), we set the hidden dimension as 16, the learning rate as 0.05. For all the backbone GNNs, their weight decay is set as 0.0005. For the augmenter GNN, its weight decay is searched in $\{0.005, 0.0005, 0.00005\}$ and its $\epsilon$ is set as 0.5. We are still going through our internal review process for releasing the code, and we expect to be able to release it before the conference.

## B    ADDITIONAL EXPERIMENTAL RESULTS

The ablation study results with SGC and APPNP backbones are presented in Table 6 and 7. Our best model ALT local obtains consistent advantages. The results are consistent with the ones presented in the main content whose backbone GNNs are GCN.

Table 6: Results of ablation study (Backbone GNN: SGC).

| Backbone | Version | Chameleon | Squirrel | Film | Computers | Photos | CS |
|---|---|---|---|---|---|---|---|
|  | None | 58.4±0.6 | 36.8±0.4 | 27.3±0.1 | 69.1±0.4 | 86.2±0.4 | 89.7±0.1 |
|  | Global | 58.5±0.9 | 37.6±0.4 | 28.6±0.2 | 69.5±0.5 | 87.2±0.2 | 90.1±0.3 |
| SGC | Local-low | 60.8±1.5 | 40.6±0.5 | 29.5±0.2 | 76.4±0.2 | 87.9±0.2 | 90.8±0.3 |
|  | Local-concat | 45.1±3.5 | 30.0±1.3 | 32.8±0.2 | 75.6±2.0 | **88.6±1.2** | 90.2±0.6 |
|  | Local | **61.9±0.8** | **42.2±1.0** | **32.9±0.2** | **79.9±0.3** | **88.6±1.5** | **92.8±0.1** |

---

[1] https://pytorch.org/

[2] https://pytorch-geometric.readthedocs.io/en/latest/

[3] https://pytorch-geometric.readthedocs.io/en/latest/modules/datasets.html

Table 7: Results of ablation study (Backbone GNN: APPNP).

| Backbone | Version | Chameleon | Squirrel | Film | Computers | Photos | CS |
|---|---|---|---|---|---|---|---|
| APPNP | None | 43.0±0.9 | 24.2±0.4 | 28.7±0.3 | 66.7±1.1 | 83.4±1.2 | 87.8±0.1 |
| | Global | 45.9±0.5 | 28.5±0.6 | 31.1±0.3 | 70.1±0.6 | 84.9±1.2 | 89.3±0.1 |
| | Local-low | 51.0±1.3 | 30.0±1.2 | 32.2±1.4 | 74.3±1.2 | 87.3±0.4 | 91.8±0.2 |
| | Local-concat | 43.2±0.4 | 26.0±0.3 | **33.8±0.9** | 73.6±2.1 | 82.5±2.1 | 89.4±0.5 |
| | Local | **55.0±0.7** | **33.5±1.7** | 33.6±0.5 | **77.6±0.8** | **88.3±0.7** | **92.4±0.4** |

We provide a hyperparameter sensitivity study as follows. Specifically, we study the sensitivity of ALT-local concerning the amplitude of the high-pass filter for the augmenter GNN (i.e, $\epsilon$ from Eq. 5b). We select GCN (Kipf & Welling, 2017) and GPRGNN (Chien et al., 2021) as backbone GNNs and conduct experiments over Cora (Yang et al., 2016), Citeseer (Yang et al., 2016), Chameleon (Rozemberczki et al., 2021), Squirrel (Rozemberczki et al., 2021) datasets. Results are presented in Figure 3 from which we observe that the model performance is stable for the selection of $\epsilon$ over four datasets and two selections of the backbone GNNs (i.e., GCN and GPRGNN).

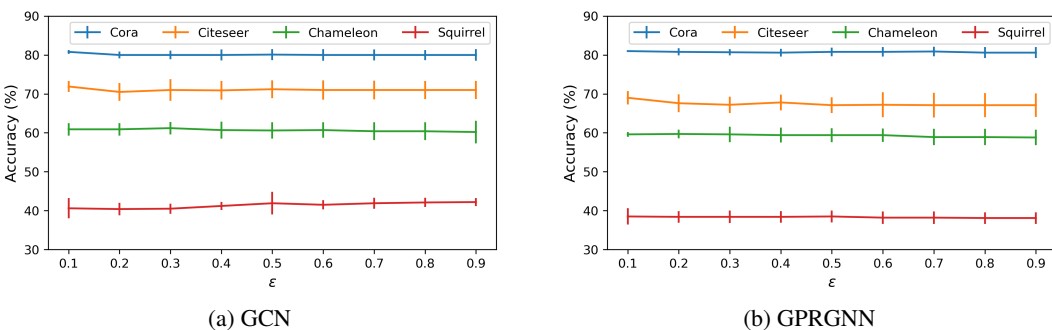

(a) GCN          (b) GPRGNN

Figure 3: Hyperparameter sensitivity of ALT with backbone GNN as (a) GCN and (b) GPRGNN.

## C    PROOF OF PROPOSITION 3

**Proposition** 3. A family of GA-MLPs are global filters if their full frequency profiles are in the form of $\mathbf{C} = \sum_k a_k \tilde{\mathbf{A}}^k + b\mathbf{I}$ which only contains non-zero diagonal entries.

*Proof.* Since $\{\tilde{\mathbf{A}}^k\}$ and $\mathbf{I}$ share the same eigenvectors, the diffusion matrix can be decomposed as

$$\mathbf{C} = \sum_k a_k \tilde{\mathbf{A}}^k + b\mathbf{I} = \mathbf{U}(\sum_k a_k (\mathbf{I} - \tilde{\mathbf{\Lambda}})^k + b\mathbf{I})\mathbf{U}^\top.$$

Hence, the frequency profile is $\mathbf{\Phi} = \sum_k a_k (\mathbf{I} - \tilde{\mathbf{\Lambda}})^k + b\mathbf{I}$ whose off-diagonal entries are zero. $\quad\square$

## D    PROOF OF LEMMA 2

**Lemma** 2 By re-weighting the edge weights non-uniformly (i.e., if re-weighting by $\mathbf{W} \odot \tilde{\mathbf{A}}$, $\exists i, j, k, l, \mathbf{W}[i, j] \neq \mathbf{W}[k, l]$), the off-diagonal entries of $\mathbf{\Phi}$ can be non-zero.

*Proof.* We follow the assumption mentioned in the proof of Lemma 1. The diffusion matrix $\mathbf{C}$ can be decomposed as $\mathbf{C} = \mathbf{U}\mathbf{\Phi}\mathbf{U}^\top$. For the full frequency profile $\mathbf{\Phi}$, its off-diagonal entry $\mathbf{\Phi}[i, j] = \sum_{l,k} \mathbf{U}[l, i]\mathbf{C}[l, k]\mathbf{U}[k, j] = 0, \forall i \neq j$. If we re-weight the diffusion matrix by $\mathbf{W} \odot \mathbf{C}$ such that $\mathbf{W}[l, k] = w_{lk}$ and $\mathbf{W}[i, j] = w \neq w_{lk}, \forall i \neq l \; and \; j \neq k$. In other words, we start from the most basic case where only one edge $(l, k)$ is re-weighted by $w_{lk}$ and all the remaining edges are re-weighted as $w$. Recall that $\mathbf{\Phi}[i, i] = \sum_{l,k} \mathbf{U}[l, i]\mathbf{C}[l, k]\mathbf{U}[k, i]$ which can be non-zero. In other

words, it is common that $\mathbf{U}[k,i] \neq 0$. Therefore, it should be easy to find a pair of node $i$ and $j$ such that $\mathbf{U}[l,i]\mathbf{U}[k,j] \neq 0$ and we obtain

$$
\begin{aligned}
\mathbf{\Phi}_{\texttt{re-weighted}}[i,j] &= (\mathbf{U}^\top (\mathbf{W} \odot \mathbf{C})\mathbf{U})[i,j] \\
&= (\mathbf{U}^\top (\mathbf{W} \odot \mathbf{C})\mathbf{U})[i,j] - w\mathbf{\Phi}[i,j] \\
&= \mathbf{U}[l,i]\mathbf{C}[l,k]\mathbf{U}[k,j](w_{lk} - w) \neq 0.
\end{aligned}
$$

Therefore, we proved that if the edge weights are re-weighted non-uniformly, the off-diagonal entries of $\mathbf{\Phi}$ can be non-zero, i.e., the GNN can be a local filter. $\qquad\square$

## E  ANALYSIS OF ALT-GLOBAL FROM THE GRAPH DIFFUSION EQUATION (GDE) PERSPECTIVE

As we claimed in Lemma 1, our proposed ALT-global can be an adaptive filter even if the given backbone GNNs only have fixed filters. Here, we prove this from the Graph Diffusion Equation (GDE) (Newman, 2018) perspective. Our proof will focus on the case where the diffusion matrix is the normalized adjacency matrix $\tilde{\mathbf{A}} = \mathbf{D}^{-\frac{1}{2}}\mathbf{A}\mathbf{D}^{-\frac{1}{2}}$ whose convolution kernel is fixed. Other cases can be proved in similar ways.

Given graph signals $\mathbf{H}$, its diffusion process can be presented as $\mathbf{H}^{(t+1)} = \tilde{\mathbf{A}}\mathbf{H}^{(t)}$. Thus, we have

$$
\mathbf{H}^{(t+1)} - \mathbf{H}^{(t)} = \frac{\mathbf{H}^{(t+1)} - \mathbf{H}^{(t)}}{(t+1) - t} = \tilde{\mathbf{A}}\mathbf{H}^{(t)} - \mathbf{H}^{(t)}. \tag{7a}
$$

In the GNN case, $t > 0$ denotes the GNN depth and in the GDE context, it denotes the diffusion time. Thus, if we set the time interval as $\Delta t$, the graph diffusion dynamics can be presented as follows,

$$
\frac{\mathbf{H}^{(t+1)} - \mathbf{H}^{(t)}}{\Delta t} = \tilde{\mathbf{A}}\mathbf{H}^{(t)} - \mathbf{H}^{(t)}, \quad \frac{d\mathbf{H}^{(t)}}{dt} = -\mathbf{L}\mathbf{H}^{(t)}, \tag{8a}
$$

where $\mathbf{L} = \mathbf{I} - \mathbf{D}^{-\frac{1}{2}}\mathbf{A}\mathbf{D}^{-\frac{1}{2}}$ is the normalized Laplacian matrix. As ALT-global re-weights all the edges into $w\tilde{\mathbf{A}}$ and $(1-w)\tilde{\mathbf{A}}$, we have

$$
\frac{d\mathbf{H}_1^{(t)}}{dt} = w\tilde{\mathbf{A}}\mathbf{H}_1^{(t)} - \mathbf{H}_1^{(t)} = (w\tilde{\mathbf{A}} - w\mathbf{I} - (1-w)\mathbf{I})\mathbf{H}_1^{(t)} = (-w\mathbf{L} - (1-w)\mathbf{I})\mathbf{H}_1^{(t)}, \tag{9a}
$$

$$
\frac{d\mathbf{H}_2^{(t)}}{dt} = (1-w)\tilde{\mathbf{A}}\mathbf{H}_2^{(t)} - \mathbf{H}_2^{(t)} = (-(1-w)\mathbf{L} - w\mathbf{I})\mathbf{H}_2^{(t)}, \tag{9b}
$$

Recap that the prediction matrix of ALT-global is by combining signals from dual backbone GNNs and an offset MLP as $\mathbf{Z} = \texttt{softmax}(\mathbf{H}_1 - \mathbf{H}_2 + \eta\mathbf{H}_{\texttt{offset}})$. We keep the assumption that the dual backbone GNNs are both GA-MLPs (Chen et al., 2021) which shares parameters with our offset MLP. Thus, we have $\mathbf{H}_1^{(0)} = \mathbf{H}_2^{(0)} = \mathbf{H}_{\texttt{offset}} = \mathbf{H} = \texttt{MLP}(\mathbf{X})$

As we are analyzing its diffusion dynamics, there is no interaction between any two columns of the feature matrix $\mathbf{H}_1^{(t)}$ (and $\mathbf{H}_2^{(t)}$). Hence, for brevity, we only show analysis of a single feature $\mathbf{h}_1^{(t)} = \mathbf{H}_1^{(t)}[:,m]$, $\mathbf{h}_2^{(t)} = \mathbf{H}_2^{(t)}[:,m]$, $\mathbf{h} = \mathbf{h}_{\texttt{offset}} = \mathbf{H}_{\texttt{offset}}[:,m]$, $\mathbf{z}^{(t)} = \mathbf{Z}^{(t)}[:,m]$, $\forall m \in \{1,\ldots,n\}$. The dual GNNs' GDEs can be presented as follows,

$$
\frac{d\mathbf{h}_1^{(t)}}{dt} = (-w\mathbf{L} - (1-w)\mathbf{I})\mathbf{h}_1^{(t)}, \tag{10a}
$$

$$
\frac{d\mathbf{h}_2^{(t)}}{dt} = (-(1-w)\mathbf{L} - w\mathbf{I})\mathbf{h}_2^{(t)}, \tag{10b}
$$

**Proposition 4.** *The solutions of Eq. 10a and Eq. 10b can be presented as*
$\mathbf{h}_1^{(t)} = \sum_{i=0}^n \left( a_i^{(0)} e^{-(w\lambda_i + (1-w))t} \right)\mathbf{u}_i$ *and* $\mathbf{h}_2^{(t)} = \sum_{i=0}^n \left( a_i^{(0)} e^{-((1-w)\lambda_i + w)t} \right)\mathbf{u}_i$, *where* $\mathbf{u}_i$ *and*
$\lambda_i$ *refers to the $i$-th eigenvector and eigenvalue of $\mathbf{L}$; initial state $a_i^{(0)}$ is determined by $\mathbf{h}_1^{(0)} = \mathbf{h}_2^{(0)} = \sum_i a_i^{(0)}\mathbf{u}_i$.*

*Proof.* Here we prove the solution of Eq. 10a and for Eq. 10b its solution can be obtained in a similar way. For Eq. 10a, by decomposing the graph signal with the eigenvectors ($\{\mathbf{u}_i\}$) of the normalized Laplacian $\mathbf{L}$ we have:

$$\mathbf{h}_1^{(t)} = \sum_i a_i^{(t)} \mathbf{u}_i. \tag{11}$$

As only $\mathbf{h}$ and $a_i$ are the functions of $t$, based on the fact that $\mathbf{L}\mathbf{u}_i = \lambda_i \mathbf{u}_i$ and $\mathbf{I}\mathbf{u}_i = \mathbf{u}_i$ we have:

$$\sum_i (\frac{da_i^{(t)}}{dt} + w\lambda_i a_i^{(t)} + (1-w)a_i^{(t)})\mathbf{u}_i = 0. \tag{12}$$

As all the eigenvectors are orthogonal with each other, by multiplying both sides of the above equation with $\mathbf{u}_i^\top$ we have

$$(\frac{da_i^{(t)}}{dt} + w\lambda_i a_i^{(t)} + (1-w)a_i^{(t)})\mathbf{u}_i = 0. \tag{13}$$

$$\frac{da_i^{(t)}}{dt} + w\lambda_i a_i^{(t)} + (1-w)a_i^{(t)} = 0. \tag{14}$$

Hence, the graph signal $\mathbf{h}_1^{(t)}$ can be represented as

$$\mathbf{h}_1^{(t)} = \sum_{i=0}^{n} \left( a_i^{(0)} e^{-(w\lambda_i + (1-w))t} \right) \mathbf{u}_i. \tag{15a}$$

Similarly, the graph signal $\mathbf{h}_2^{(t)}$ can be presented as

$$\mathbf{h}_2^{(t)} = \sum_{i=0}^{n} \left( a_i^{(0)} e^{-((1-w)\lambda_i + w)t} \right) \mathbf{u}_i. \tag{16a}$$

$\square$

Thus, aggregated signal can be presented as (here we use $\mathbf{h}_{\texttt{offset}} = \mathbf{h}^{(0)} = \sum_{i=1}^{n} a_i^{(0)}\mathbf{u}_i$)

$$\mathbf{z}^{(t)} = \mathbf{h}_1^{(t)} - \mathbf{h}_2^{(t)} + \eta\mathbf{h}_{\texttt{offset}} \tag{17a}$$

$$= \sum_{i=0}^{n} a_i^{(0)} \left( e^{-(w\lambda_i + (1-w))t} - e^{-((1-w)\lambda_i + w)t} + \eta \right) \mathbf{u}_i \tag{17b}$$

According to the graph signal processing Shuman et al. (2013), $\mathbf{u}_i$ denotes the graph signal with $\lambda_i$ frequency. Hence, $a_i^{(0)} \left( e^{-(w\lambda_i + (1-w))t} - e^{-((1-w)\lambda_i + w)t} + \eta \right)$ denotes the amplitude of the the $\lambda_i$-frequency signal after filtered by ALT-global. We know the signal before filtering (i.e., diffusion) is

$$\mathbf{h}^{(0)} = \mathbf{h}_1^{(0)} = \mathbf{h}_2^{(0)} = \mathbf{h}_{\texttt{offset}}^{(0)} = \sum_{i=0}^{n} a_i^{(0)}\mathbf{u}_i, \tag{18}$$

and the amplitude of the the $\lambda_i$-frequency signal before filtering is $a_i^0$. Hence, the filter response to $\lambda_i$ frequency is

$$\Phi(\lambda_i) = \frac{a_i^{(0)} \left( e^{-(w\lambda_i + (1-w))t} - e^{-((1-w)\lambda_i + w)t} + \eta \right)}{a_i^{(0)}} \tag{19a}$$

$$= e^{-(w\lambda_i + (1-w))t} - e^{-((1-w)\lambda_i + w)t} + \eta \tag{19b}$$

It is clear when $w > 0$,, $\Phi(\lambda_i)$ is a monotonically decreasing function and when $w < 0$, $\Phi(\lambda_i)$ is a monotonically increasing function. With appropriate $\eta$ and different $w$, ALT-global can be instantiated as either a low-pass filter or a high-pass filter.

