# OpenReview forum: "Node Classification Beyond Homophily: Towards a General Solution"
_ICLR.cc/2023/Conference — Submitted to ICLR 2023_

### Official Review · Reviewer_Jqz9 · 2022-10-24

**Confidence:** 5
**Correctness:** 1
**Technical Novelty And Significance:** 3
**Empirical Novelty And Significance:** 4
**Recommendation:** 6

**Clarity, Quality, Novelty And Reproducibility:**

* Clarity in the spectral stuff could be improved but in general it is well written.
* Novelty. Seems that is not incremental wrt FAGCN but I am open to discuss it.
* No code released yet but experimental protocol specified.

**Details Of Ethics Concerns:**

Ok.

**Strength And Weaknesses:**

* Strengths: results are very challenging and
* Weaknesses: the underlying theory could be clarified more clearly for the nonexperts in spectral graph theory.

**Summary Of The Paper:**

Main concern. How to accommodate GNNs to deal naturally with heterophily and provide a theoretical analysis (from signal processing perspective). The core idea is to transform/combine low-pass filter into/with high-pass, for instance. There are two learnable parameters: w and \eta whose values lead to low-pass or high-pass filters (adaptive filters):

The above conditions are sufficient and in fact, there are many other combinations of w and η that can produce low-pass/high-pass filters
As stated in the proof of Lemma 1. I found this idea very interesting, in particular the distinction between GLOBAL and LOCAL filtering. In particular, the insight of how GLOBAL filtering degrades the performance in a heterophilic scenario is useful.

Related work is analyzed and the contribution evolves from FAGCN which explicitly mixes high-frequency and low-frequency signals. ALT generalizes this idea to the ‘mixture of complementary filters’; thus, even though the backbone GNN’s convolution kernel is unknown

Results are very challenging since for heterophilic graphs, the proposed approach ALT increases the performance and for homophilic case, it preserves it or even improves it. The ablation study is very useful to understand the contribution of the local filter.

Notes. Please, clarify that the frequency profile is basically the function phi applied on the spectrum (e.g phi = exp() when we apply a heat kernel) under the spectral theorem: phi(A) = U*phi(lambda)*U^T.



**Summary Of The Review:**

The paper provides nice results regarding how to minimally modify existing GNNs to deal with heterophilic graphs. The theory could be better explained but the paper is in general readable and direct.

---

> ### Author Response · Authors · 2022-11-12
> **Response to reviewer Jqz9**
>
> We thank your careful review!
>
> For your question,
>
> **Q1.** Please, clarify that the frequency profile is basically the function phi applied on the spectrum (e.g phi = exp() when we apply a heat kernel) under the spectral theorem: phi(A) = $U*phi(lambda)*U^T$.
>
> **A1.** We are sorry that you are confused about the concept of frequency profile. For a given graph convolution kernel $f(\lambda)$, the graph convolution can be presented as $\mathbf{y}=\mathbf{U}\mathbf{G}\mathbf{U}^{\top}\mathbf{x}$, where $\mathbf{G}=\texttt{diag}(f(\lambda_i)_{i=1}^{n})$
>
> From the spatial perspective, the convolution is equivalent to $\mathbf{y}=\mathbf{U}\mathbf{G}\mathbf{U}^{\top}\mathbf{x}=\mathbf{C}\mathbf{x}$, where $\mathbf{C}$ is the diffusion matrix. Hence, according to the definition of frequency profile [1], $\Phi_{fp}=\texttt{diag}^{-1}(\mathbf{U}^{\top}\mathbf{C}\mathbf{U})=\texttt{vector}({f(\lambda_i)}_{i=1}^{n})$ which is exactly the convolution kernel on every eigenvalue (frequency), i.e., filter.
>
> Let us know if you are still confused about this concept.
>
> [1] Balcilar, Muhammet, et al. "Analyzing the expressive power of graph neural networks in a spectral perspective." ICLR, 2021.

---

> > ### Comment · Reviewer_Jqz9 · 2022-11-14
> > **Answer**
> >
> > It is basically what I understood but vectorized

---

### Official Review · Reviewer_AcUh · 2022-10-24

**Confidence:** 4
**Correctness:** 3
**Technical Novelty And Significance:** 2
**Empirical Novelty And Significance:** 2
**Recommendation:** 3

**Clarity, Quality, Novelty And Reproducibility:**

Clarity: medium
Quality: medium
Novelty: low
Reproducibility: NA

**Strength And Weaknesses:**

## Strength
The frequency analysis is somehow interesting.

## Weaknesses
1. The writing needs to be improved.
2. The idea is not novel.


## Questions and Comments

1. “our core idea is to adaptively combine signals from two filters with the complementary filter characteristics. For example, if a low-pass filter GNN is given, it should be adaptively combined with another high-pass filter.” Combining low-pass filter and high-pass filter adaptively is exactly the same as [3] and as the author mentioned, “equip GNNs with an adaptive filter” is not a new idea.

2. The authors show that “The filter characteristic of the proposed ALT-global (Eq. 1d) is adaptive…”. How does the learned filter compared with GPRGNN and BernNet?

3. “applies a global filter to every node, which could lead to suboptimal performance.” Any evidence to this claim?

4. The ALT-LOCAL is very similar to FB-GAT, which combines low-pass filter, high-pass filter and learnable edge weights adaptively.

5. It is invalid to use validation label in the training process.

5. Some recent works find some empirical and theoretical  evidence that heterophily is not always harmful and homophily is not always necessary for GNNs [1,2,3]. How.does this work align with the previous works.

6. Some missing comparison, e.g. LINKX [4], BernNet [5], ACM-GCN [3] and GloGNN[6].

[1] Zhu J, Yan Y, Zhao L, et al. Beyond homophily in graph neural networks: Current limitations and effective designs[J]. Advances in Neural Information Processing Systems, 2020, 33: 7793-7804.

[2] Ma Y, Liu X, Shah N, et al. Is homophily a necessity for graph neural networks?[J]. arXiv preprint arXiv:2106.06134, 2021.

[3] Luan S, Hua C, Lu Q, et al. Is Heterophily A Real Nightmare For Graph Neural Networks To Do Node Classification?[J]. arXiv preprint arXiv:2109.05641, 2021.

[4] Lim D, Hohne F, Li X, et al. Large scale learning on non-homophilous graphs: New benchmarks and strong simple methods[J]. Advances in Neural Information Processing Systems, 2021, 34: 20887-20902

[5]  He M, Wei Z, Xu H. Bernnet: Learning arbitrary graph spectral filters via bernstein approximation[J]. Advances in Neural Information Processing Systems, 2021, 34: 14239-14251.

[6] Li X, Zhu R, Cheng Y, et al. Finding Global Homophily in Graph Neural Networks When Meeting Heterophily[J]. arXiv preprint arXiv:2205.07308, 2022.

[7] Eli Chien, Jianhao Peng, Pan Li, and Olgica Milenkovic. Adaptive universal generalized pagerank
graph neural network. In International Conference on Learning Representations, 2021.

[8] Luan S, Zhao M, Hua C, et al. Complete the missing half: Augmenting aggregation filtering with diversification for graph convolutional networks[J]. arXiv preprint arXiv:2008.08844, 2020.



**Summary Of The Paper:**

The authors ALT to handle graphs with either low or high homophily by decomposing a given graph into two components, extracting complementary graph signals from these two components, and adaptively merge the graph signals for node classification.

**Summary Of The Review:**

Although some theoretical analysis is interesting, this paper lacks novelty and strong experimental support. Thus, I don't think it is ready to be published.

---

> ### Author Response · Authors · 2022-11-12
> **Response to reviewer AcUh (1/2)**
>
> We thank your careful review! You mentioned some related works which are missing in our paper and we just cited them in the revised version (Section 5). Our following responses to your questions will involve many references which are put in the end.
>
> **Q1.** Combining low-pass filter and high-pass filter adaptively is exactly the same as [3] and as the author mentioned, “equip GNNs with an adaptive filter” is not a new idea.
>
> **A1.** We agree with the reviewer's point that 'equipping GNNs with an adaptive filter is not a new idea'. The comparison between our work and a representative (adaptive filter) work FAGCN is presented in the 'Remarks' paragraph at the end of Section 3.4.
>
> **Q2.** The authors show that “The filter characteristic of the proposed ALT-global (Eq. 1d) is adaptive…”. How does the learned filter compare with GPRGNN and BernNet?
>
> **A2.** Thank you for introducing the BernNet paper [4]. Based on our understanding this paper sets constraints on the learned filters (to follow the Bernstein basis) to ensure it learns well-posed filters. This is an interesting and valuable work. For your question, based on our definition of the "global filters vs. local filters" (Section 3.3), **both GPRGNN [8] and BernNet [4] should be categorized into the global filters** because they both work on the diagonal entries of the Lambda matrix and cannot modulate the input graph signals. Our ALT-global is guaranteed to be **as spectral expressive as** GPRGNN and BernNet, and our ALT-local is guaranteed to be **more spectral expressive** than GPRGNN and BernNet, even the plugged-in backbone GNN is with a fixed filter.
>
> **Q3.** “applies a global filter to every node, which could lead to suboptimal performance.” Any evidence to this claim?
>
> **A3.** There are some works [6][7] mention that **the heterophily patterns differ between different node pairs**. A straightforward example is that a graph is combined from a homophilic graph and a heterophilic graph (e.g., heterosexual and homosexual dating graphs with a common bisexual node). In that case, empirically, an optimal GNN should work as a low-pass filter in the homophilic part but it should work as a high-pass filter in the heterophilic part. Thus, A global (e.g., high/low-pass) filter is suboptimal.
>
> **Q4.** The ALT-LOCAL is very similar to FB-GAT, which combines a low-pass filter, a high-pass filter, and learnable edge weights adaptively.
>
> **A4.** The reviewer's point is right. Combining two filters locally is not a  new idea. The comparison between our work and the representative work FAGCN (in the 'Remarks' paragraph at the end of Section 3.4.) should be helpful to better clarify the contribution of this paper. In addition, could you provide the paper name of FB-GAT so that we can check it in detail?
>
> **Q5.** It is invalid to use validation label in the training process.
>
> **A5.** As this concern is also mentioned by other reviewers, we answer it in the general response "Regarding the validity of the training strategy" part and hope it can address your concern. Thank you!
>
> **Q6.** Some recent works find some empirical and theoretical evidence that heterophily is not always harmful and homophily is not always necessary for GNNs [1,2,3]. How does this work align with the previous works.
>
> **A6.** This is an interesting question and we want to share some insights here. For your question, "the connection between the works you mentioned and our developed theory", the answer is, our work is orthogonal with existing theory (mainly from [5]) on understanding GNN node classification. Our analysis is a step forward based on the classic graph signal processing (i.e., **more focus is on the diffusion matrix**) to show that our framework can handle a more complicated topology and node feature distribution (e.g., graph with various local heterophily). As a comparison, the core idea from [5] tells us that if graph topology and node feature follow **certain distributions**, **a simple diffusion matrix (e.g., the normalized adjacency matrix) is enough** to get good performance. We also noticed that the "post-aggregation node similarity matrix" from [2] is an interesting metric and it provides a fast tool to verify the quality of the diffusion matrix. How to use the "post-aggregation matrix" in a data-driven way is worth studying.

---

> > ### Author Response · Authors · 2022-11-12
> > **Response to reviewer AcUh (2/2)**
> >
> > **Q7.** Some missing comparison, e.g. LINKX [4], BernNet [5], ACM-GCN [3] and GloGNN[6].
> >
> > **A7.** Thanks for mentioning these related works. We cited them in our revised version. For comparison, we use the reported performance of LINKX [1], ACM-GCN [2], and GloGNN [3] from the latest ACM-GCN paper [2] and implement our ALT model and BernNET [4] in the same setting (48/32/20% data split) as [2] reported. We implement our ALT with APPNP [9] (a fixed filter GNN) as the backbone. The results are as Table 1 shows, from which we observe that (1) our **ALT-APPNP can get comparable performance** against state-of-the-art methods on most of the datasets (except Chameleon and Squirrel) and (2) LINKX, ACM-GCN++, and GloGNN++ all used a trick to encode adjacency matrix (i.e., $\texttt{MLP}(\mathbf{A})$) as a supplementary of node embedding. We found this trick is independent of the model design so we added this trick into our model and **the enhanced ALT-APPNP+ get very strong performance** on Chameleon and Squirrel datasets.
> >
> >
> > |                 | Cornell                | Wisconsin          | Texas              | Film               | Chameleon          | Squirrel           | Cora               | Citeseer       | PubMed         |
> > |-----------------|------------------------|--------------------|--------------------|--------------------|--------------------|--------------------|--------------------|--------------------|--------------------|
> > | **ACM-GCN**     |     85.14$\pm$6.07     | 88.43$\pm$3.22     | 87.84$\pm$4.4      | 36.63$\pm$0.84     | 69.14$\pm$1.91     | 55.19$\pm$1.49     | 87.91$\pm$0.95     | 77.32$\pm$1.7      | 90.00$\pm$0.52     |
> > | **BernNet**     | 81.05$\pm$8.39         | 87.29$\pm$4.63     | 82.58$\pm$4.92     | 34.16$\pm$1.46     | 45.38$\pm$1.91     | 33.12$\pm$1.36     | 87.62$\pm$0.56     | 76.09$\pm$0.31     | 86.21$\pm$0.29     |
> > | **LINKX**       |        77.84$\pm$5.81  | 75.49$\pm$5.72     | 74.60$\pm$8.37     | 36.10$\pm$1.55     | 68.42$\pm$1.38     | 61.81$\pm$1.80     | 84.64$\pm$1.13     | 73.19$\pm$0.99     | 87.86$\pm$0.77     |
> > | **ACMII-GCN++** | 86.49$\pm$6.73         | 88.43$\pm$3.66     | 88.38$\pm$3.43     | 37.09$\pm$1.32     | 74.76$\pm$2.2      | 67.4$\pm$2.21      | 88.25$\pm$0.96     | 77.12$\pm$1.58     | 89.71$\pm$0.48     |
> > | **GloGNN++**    |        85.95$\pm$5.10  | 88.04$\pm$3.22     | 84.05$\pm$4.90     | **37.70$\pm$1.40** | 71.21$\pm$1.84     | 57.88$\pm$1.76     | 88.33$\pm$1.09     | 77.22$\pm$1.78     | 89.24$\pm$0.39     |
> > | **ALT-APPNP**   | 86.84$\pm$4.30         | **88.89$\pm$2.45** | 88.72$\pm$3.28     | 37.58$\pm$0.67     | 66.74$\pm$2.03     | 54.29$\pm$1.17     | 88.09$\pm$0.46     | 77.61$\pm$1.46     | 89.92$\pm$0.64     |
> > | **ALT-APPNP+**  | **90.35$\pm$4.47**     | 88.60$\pm$3.33     | **89.47$\pm$2.15** | 37.28$\pm$1.22     | **77.02$\pm$1.86** | **69.39$\pm$1.47** | **89.56$\pm$1.25** | **79.86$\pm$1.18** | **90.32$\pm$0.47** |
> >
> > *Table 1. Performance comparison with SOTA methods (mean$\pm$std)*
> >
> > [1] Lim D, Hohne F, Li X, et al. Large scale learning on non-homophilous graphs: New benchmarks and strong simple methods[J]. NeurIPS, 2021.
> >
> > [2] Luan, Sitao, et al. "Revisiting Heterophily For Graph Neural Networks." arXiv preprint arXiv:2210.07606 (2022).
> >
> > [3] Li X, Zhu R, Cheng Y, et al. Finding Global Homophily in Graph Neural Networks When Meeting Heterophily[J]. arXiv preprint arXiv:2205.07308, 2022.
> >
> > [4] He M, Wei Z, Xu H. Bernnet: Learning arbitrary graph spectral filters via bernstein approximation[J]. NeurIPS, 2021.
> >
> > [5] Ma Y, Liu X, Shah N, et al. Is homophily a necessity for graph neural networks?[J]. arXiv preprint arXiv:2106.06134, 2021.
> >
> > [6] Jiong Zhu, Ryan A Rossi, Anup Rao, Tung Mai, Nedim Lipka, Nesreen K Ahmed, and Danai Koutra. Graph neural networks with heterophily. AAAI, 2021a.
> >
> > [7] Wang, Tao, et al. "Powerful graph convolutional networks with adaptive propagation mechanism for homophily and heterophily." AAAI 2022.
> >
> > [8] Eli Chien, Jianhao Peng, Pan Li, and Olgica Milenkovic. Adaptive universal generalized pagerank graph neural network. ICLR, 2021.
> >
> > [9] Gasteiger, Johannes, Aleksandar Bojchevski, and Stephan Günnemann. "Predict then Propagate: Graph Neural Networks meet Personalized PageRank." ICLR. 2019.

---

### Official Review · Reviewer_j7Qg · 2022-10-26

**Confidence:** 4
**Clarity, Quality, Novelty And Reproducibility:** The paper is clearly written but the …
**Correctness:** 3
**Technical Novelty And Significance:** 1
**Empirical Novelty And Significance:** 1
**Recommendation:** 3

**Strength And Weaknesses:**

# Strength:

1. The paper is well-written and easy to follow.

2. The proposed idea can be used as a general framework to potentially improve any GNN models.

3. The motivation of ALT-Global is clear and valid.

# Weakness:

1. The novelty of the proposed idea is limited. The idea of combining graph signals from different graph filters is implicitly encoded in polynomial filters of graph signals, and this is a well-studied problem in the literature. For instance, in the analysis of ALT-Global (Section 3.2), the paper assumes the GA-MLP architecture: $H=C~MLP(X)$ and all three MLPs share the same parameters. Then the proposed ALT essentially reduces to a specific case of GPRGNN [1] whose polynomial filter is even more flexible and adaptive.

2. The proposed framework might be able to improve some classic GNN backbones (such as GCN, SGC, and APPNP) that intrinsically adopt low-pass filters and therefore can not handle heterophilic data. However, it is unclear how the proposed idea improves stronger approaches such as GPRGNN [1] and other approaches with high-frequency filters.

3. In the ablation study (Table 3), even for GCN, the ALT-Glocal variant only improves the performance very marginally. I suspect that ALT-Global can not improve the performance of GPRGNN, APPNP, or GCNII. The paper only presents the ablation study on GCN, but more results are needed to clarify this concern.

4. The local adaptive method proposed in Section 3.4 is heuristic and ad-hoc. The training procedure in Section 3.5 seems to be computationally expensive but there is no discussion on the running time and training cost.

5. There is a lack of comparison with other local adaptive approaches which also adaptively adjust the edge weight as in ALT-Local. For instance, graph attention networks [2] adjust the weight by attention scores; ElasticGNN [3] models the locally adaptive smoothness over the graph by graph trend filtering; DAGNN [4] adjusts the feature combination from different propagation layers by attention scores; More discussion and comparison will be helpful to justify the advantages of the proposed idea. Overall, it will be beneficial to show how the proposed idea improves state-of-art algorithms instead of old algorithms.

[1] Adaptive Universal Generalized PageRank Graph Neural Network, ICLR 2021

[2] Graph Attention Networks, ICLR 2018

[3] Elastic Graph Neural Networks, ICML 2021

[4] Towards Deeper Graph Neural Networks, KDD 2020




**Summary Of The Paper:**

The paper proposes a general framework ALT to deal with the heterophily problem in GNNs. The main idea is to design multiple GNNs or MLP with different graph filters and it finally merges their predictions regardless of the GNN backbones. A local version is also developed for further improvement.

**Summary Of The Review:**

The paper proposes a general framework ALT to deal with the heterophily problem in GNNs. The main idea is to design multiple GNNs or MLP with different graph filters and it finally merges their predictions regardless of the GNN backbones. The novelty is limited compared with existing approaches, and it is unclear how it advances the state-of-art algorithms. Moreover, a local version is developed for further improvement but the proposed method is quite heuristic without clear justification and running time analysis.

---

> ### Author Response · Authors · 2022-11-13
> **Response to reviewer j7Qg (1/2)**
>
> We thank your careful review and comments! Our following responses to your questions/comments will involve many references which are put in the end.
>
> **Q1.** The idea of combining graph signals from different graph filters is implicitly encoded in polynomial filters of graph signals, and this is a well-studied problem in the literature. The proposed ALT essentially reduces to a specific case of GPRGNN whose polynomial filter is even more flexible and adaptive.
>
> **A1.** We do agree with the reviewer's point that combining graph filters is a well-studied problem. However, as far as our knowledge, our work is the first work to generalize the idea of 'combining a low-pass filter and a high-pass filter' to the idea of **'combining two complementary filters'**, which empowers the flexibility of our framework. For our assumption on the GA-MLP architecture [1], as you mentioned, it is only for analysis purposes. In addition, we want to clarify that **GPRGNN [2] is a member of GA-MLPs** so our GA-MLP-based analysis will reduce the case of GPRGNN is not true. In fact, the diffusion matrix $\mathbf{C}$ can be of any form but is not limited to the polynomial of the normalized adjacency matrix.
>
> **Q2.** it is unclear how the proposed idea improves stronger approaches such as GPRGNN and other approaches with high-frequency filters.
>
> **A2.** This is a good question. In **Proposition 3**, we present the limitation of a family of GA-MLPs which **include GPRGNN**, and that is the main motivation why we develop the advanced model ALT-local. As GPRGNN is with global filter, Section 3.3 explains the limitation of global filters and that leads to the design of our proposed ALT-local.
>
> **Q3.** The paper only presents the ablation study on GCN, but more results are needed to clarify this concern.
>
> **A3.** Thanks for the careful inspection. We provided an ablation study of GCN in the main content and **the ablation studies of SGC [3] and APPNP [4] were included in the appendix**. For your suggestions, we supplement the ablation studies with GCNII [5] and GPRGNN [2] backbones as Table 1 and Table 2 show. In general, the conclusion is consistent with the content that the ALT-local variant has the best performance.
> | Backbone | Version      | Texas            | Wisconsin        | Cornell          | Computers        | Photos           | CS               |
> |----------|--------------|------------------|------------------|------------------|------------------|------------------|------------------|
> | GPRGNN   | None         | 69.1$\pm$1.0     | 72.4$\pm$1.6     | 69.6$\pm$2.5     | 85.0$\pm$0.5     | 92.4$\pm$0.2     | 92.3$\pm$0.1     |
> |          | Global       | 69.7$\pm$2.1     | 75.5$\pm$6.8     | 69.7$\pm$4.5     | 85.6$\pm$0.8     | **92.6$\pm$0.3** | 92.6$\pm$1.3     |
> |          | Local-low    | 73.9$\pm$4.3     | 77.3$\pm$3.6     | 70.5$\pm$1.1     | 85.5$\pm$1.8     | 92.5$\pm$0.4     | 92.5$\pm$0.3     |
> |          | Local-concat | 69.8$\pm$2.6     | 71.3$\pm$1.4     | 70.0$\pm$5.0     | 70.5$\pm$0.8     | 83.5$\pm$4.1     | 87.9$\pm$1.3     |
> |          | Local        | **76.3$\pm$1.5** | **79.5$\pm$0.7** | **70.9$\pm$2.9** | **85.9$\pm$1.5** | **92.6$\pm$0.3** | **93.2$\pm$0.2** |
>
> *Table 1. Results of ablation study with GPRGNN backbone.*
>
> | Backbone | Version      | Texas            | Wisconsin        | Cornell          | Computers        | Photos           | CS               |
> |----------|--------------|------------------|------------------|------------------|------------------|------------------|------------------|
> | GCNII    | None         | 57.7$\pm$5.3     | 51.4$\pm$3.5     | 53.5$\pm$3.0     | 60.6$\pm$1.9     | 77.0$\pm$4.9     | 84.2$\pm$1.0     |
> |          | Global       | 59.8$\pm$4.2     | 59.8$\pm$5.5     | 59.2$\pm$2.6     | 65.7$\pm$2.8     | 83.7$\pm$3.4     | 86.8$\pm$1.8     |
> |          | Local-low    | 62.8$\pm$1.9     | 60.7$\pm$2.3     | 63.3$\pm$5.8     | 66.6$\pm$3.7     | 85.2$\pm$5.9     | 85.9$\pm$1.0     |
> |          | Local-concat | 64.8$\pm$2.9     | **65.3$\pm$3.0** | 68.5$\pm$2.2     | 72.8$\pm$1.4     | 74.5$\pm$5.3     | 79.5$\pm$1.7     |
> |          | Local        | **64.9$\pm$2.6** | **65.3$\pm$4.3** | **70.3$\pm$2.7** | **74.2$\pm$2.8** | **89.2$\pm$0.5** | **88.6$\pm$0.1** |
>
> *Table 1. Results of ablation study with GCNII backbone.*
>
> **Q4.** The training procedure in Section 3.5 seems to be computationally expensive but there is no discussion on the running time and training cost.
>
> **A4.** As other reviewers also show their concerns about the training process, we give **a general response on the top** which includes our response to the training cost (**"Regarding the training efficiency"** part). Hope it can address your concern on this point. Thank you!

---

> > ### Author Response · Authors · 2022-11-13
> > **Response to reviewer j7Qg (2/2)**
> >
> > **Q5.** There is a lack of comparison with other local adaptive approaches which also adaptively adjust the edge weight as in ALT-Local. For instance, graph attention networks adjust the weight by attention scores; ElasticGNN models the locally adaptive smoothness over the graph by graph trend filtering; DAGNN adjusts the feature combination from different propagation layers by attention scores; More discussion and comparison will be helpful to justify the advantages of the proposed idea. Overall, it will be beneficial to show how the proposed idea improves state-of-art algorithms instead of old algorithms.
> >
> > **A5.** Thanks for mentioning these works. However, the methods mentioned in this comment (i.e., GAT [6], ElasticGNN [7], DAGNN [8]) are **not designed for heterophilic graph settings** and they even did not include heterophilic graphs in their experimental settings. Thus, they are not the advanced methods on this topic (node classification beyond homophily). That is the reason why we do not select them in the baseline methods. And, **the baseline methods presented in our paper are not "old methods"**. For example, technically, FAGCN [9] is developed based on GAT, and FAGCN's attention weight can be negative. GPRGNN [2] and FAGCN are both proposed in recent two years. Both their performance gets improved significantly after being equipped with our proposed ALT framework which is presented in Table 1 in our main content.
> >
> > Even though pursuing state-of-the-art (SOTA) performance is not the major goal of our methods (our focus is flexibility and generality), we provide **performance comparison with SOTA methods in the "Response to reviewer AcUh 2/2"** where our method gets comparable or even stronger performance compared with SOTA methods. For brevity we do not pose repeated content here and could you check the "Response to reviewer AcUh 2/2" part to see if it addresses your concern? Thank you!
> >
> > [1] Chen, Lei, Zhengdao Chen, and Joan Bruna. "On Graph Neural Networks versus Graph-Augmented MLPs." ICLR, 2021.
> >
> > [2] Eli Chien, Jianhao Peng, Pan Li, and Olgica Milenkovic. Adaptive universal generalized pagerank graph neural network. ICLR, 2021.
> >
> > [3] Wu, Felix, et al. "Simplifying graph convolutional networks." ICML, 2019.
> >
> > [4] Gasteiger, Johannes, Aleksandar Bojchevski, and Stephan Günnemann. "Predict then Propagate: Graph Neural Networks meet Personalized PageRank." ICLR. 2019.
> >
> > [5] Chen, Ming, et al. "Simple and deep graph convolutional networks." ICML, 2020.
> >
> > [6] Graph Attention Networks, ICLR 2018.
> >
> > [7] Elastic Graph Neural Networks, ICML 2021.
> >
> > [8] Towards Deeper Graph Neural Networks, KDD 2020.
> >
> > [9] Bo, Deyu, et al. "Beyond low-frequency information in graph convolutional networks." AAAI, 2021.

---

### Official Review · Reviewer_qBZ9 · 2022-10-27

**Confidence:** 4
**Correctness:** 3
**Technical Novelty And Significance:** 4
**Empirical Novelty And Significance:** 3
**Recommendation:** 6

**Clarity, Quality, Novelty And Reproducibility:**

**Clarity**
Writing is very easy to follow.

**Quality**
Good.

**Novelty**
Good.

**Reproducibility**
Should be reproducible.

**Strength And Weaknesses:**

**Good Empirical Results**
The authors performed a very comprehensive study on a wide range of datasets. The authors also compared against a wide range of baselines. Overall the improvement is quite significant. Also, the authors conducted a detailed ablation study of ALT-local. That highlight the importance of frequency modulator.

**Well Motivated/Explained Architecture Design**
The ALT design is well motivated by the combination of two filters with complementary characteristics. The edge reweighting is well motivated by the need to switch from global filter to local filter. The augmentation GNN is also well motivated by the discriminative argument to use a high pass filter.

**Issue with Experiments**
One complain I have is that the bi-level optimization explicitly use the validation vertices to train the augmentation. This makes the empirical comparison unfair for the baselines. My suggestion would be to split part of the training nodes for training the backbone and augmentation separately.

**Summary Of The Paper:**

**Summary**
The paper proposed a method ALT to convert any graph neural network to be effective for non-homophily graphs. The key idea is to use two backbone GNNs and an additional MLP to shift (or adapt) the frequency response function of the diffusion filter. The authors also propose a more complicated version of ALT which compute a weight of edges with an additional GNN. The resulting model is optimized with a bi-level optimization method on both the train and valid vertices. ALT shows good empirical performance, especially on non-homophily graphs and on top of the classical GNNs.

**Summary Of The Review:**

Borderline accept, would be stronger if issue with experiments fixed.

---

> ### Author Response · Authors · 2022-11-12
> **Response to reviewer qBZ9**
>
> We thank your careful review!
>
> **Q1.** One complain I have is that the bi-level optimization explicitly use the validation vertices to train the augmentation
>
> **A1.** This is a valid concern and your suggestion is nice. Could you check our general response "Regarding the validity of the training strategy" part to see if it addresses your concern? Thank you!

---

### Author Response · Authors · 2022-11-12
**A General Response**

**We thank the reviewers' time, effort, and constructive suggestions.** A common concern is the training strategy's validity (reviewer qBZ9 and AcUh) and efficiency (reviewer j7Qg). Here we give a general response to this concern. **For other specific questions from every reviewer, we will respond to them as soon as possible.**

**Regarding the validity of the training strategy**, we formulate it as a bilevel optimization problem, and the graph itself (and the graph augmenter) is viewed as hyperparameters (to be tuned/learned over the validation data [4]). It is common among existing structure learning works (e.g., [1] and [2]). If the reviewers worry that this setting is not fair, we re-run a part of the experiments by jointly training the backbone classifier and augmenter over the training nodes and using the validation nodes to do early stopping (just as baselines did) and report the performance in **Table 1** and **Table 2**. We did not observe significant performance dropping by this training strategy and even surprisingly observe some performance improvement by the joint training strategy on some datasets. All the supplementary experimental results in this rebuttal/discussion will be based on the joint training strategy.

|            |         | Chameleon    | Squirrel     | Texas        | Wisconsin    | Cornell      | Film         | Cornell5     | Penn94       |
|------------|---------|--------------|--------------|--------------|--------------|--------------|--------------|--------------|--------------|
| ALT-APPNP  | Bilevel | 55.0$\pm$0.7 | 33.5$\pm$1.7 | 72.1$\pm$1.7 | 76.3$\pm$3.3 | 75.0$\pm$1.6 | 33.6$\pm$0.5 | 74.1$\pm$0.1 | 75.7$\pm$0.5 |
|            | Joint   | 54.1$\pm$0.7 | 38.2$\pm$1.0 | 71.2$\pm$2.9 | 76.6$\pm$2.7 | 78.4$\pm$3.4 | 34.0$\pm$0.3 | 73.7$\pm$0.2 | 75.8$\pm$0.3 |
| ALT-GPRGNN | Bilevel | 59.4$\pm$1.2 | 38.2$\pm$0.9 | 76.3$\pm$1.5 | 79.5$\pm$0.7 | 70.9$\pm$2.9 | 32.1$\pm$0.8 | 75.9$\pm$0.3 | 80.4$\pm$0.2 |
|            | Joint   | 59.2$\pm$1.6 | 38.5$\pm$0.7 | 75.4$\pm$2.4 | 79.7$\pm$0.5 | 70.6$\pm$1.5 | 32.8$\pm$1.0 | 76.3$\pm$0.6 | 80.3$\pm$0.1 |

*Table 1. Training strategy comparison on heterophilic graphs (mean$\pm$std)*

|            |         | Cora         | Citeseer     | Pubmed       | DBLP         | Computers    | Photos       | CS           | Physics      |
|------------|---------|--------------|--------------|--------------|--------------|--------------|--------------|--------------|--------------|
| ALT-APPNP  | Bilevel | 82.4$\pm$0.4 | 71.7$\pm$0.2 | 79.5$\pm$0.8 | 84.4$\pm$0.1 | 77.6$\pm$0.8 | 88.3$\pm$0.7 | 92.4$\pm$0.4 | 95.5$\pm$0.1 |
|            | Joint   | 82.7$\pm$0.3 | 72.1$\pm$0.3 | 79.3$\pm$0.2 | 84.2$\pm$0.4 | 87.2$\pm$1.3 | 92.5$\pm$0.4 | 93.8$\pm$0.1 | 96.4$\pm$0.1 |
| ALT-GPRGNN | Bilevel | 80.9$\pm$0.3 | 68.8$\pm$0.2 | 78.2$\pm$0.4 | 84.4$\pm$0.3 | 85.9$\pm$1.5 | 92.6$\pm$0.3 | 93.2$\pm$0.2 | 95.7$\pm$0.1 |
|            | Joint   | 83.0$\pm$0.4 | 70.0$\pm$1.2 | 79.6$\pm$0.5 | 84.4$\pm$0.1 | 85.2$\pm$0.8 | 92.9$\pm$0.2 | 92.9$\pm$0.2 | 96.2$\pm$0.1 |

*Table 2. Training strategy comparison on homophilic graphs (mean$\pm$std)*

**Regarding the training efficiency**, as the optimization objective is a standard bilevel optimization problem, there are many existing solvers (e.g., [3-6]) whose computational costs are diverse. As the efficiency of different solvers is not the contribution and focus of this paper, we only present the model complexity at the end of Section 3.5.

[1] Franceschi, Luca, et al. "Learning discrete structures for graph neural networks." ICML, 2019.

[2] Xu, Zhe, Boxin Du, and Hanghang Tong. "Graph sanitation with application to node classification." TheWebConf, 2022.

[3] Finn, Chelsea, Pieter Abbeel, and Sergey Levine. "Model-agnostic meta-learning for fast adaptation of deep networks." ICML, 2017.

[4] Franceschi, Luca, et al. "Bilevel programming for hyperparameter optimization and meta-learning." ICML, 2018.

[5] Franceschi, Luca, et al. "Forward and reverse gradient-based hyperparameter optimization." ICML, 2017.

[6] Pedregosa, Fabian. "Hyperparameter optimization with approximate gradient." ICML, 2016.

---

### Decision · Program_Chairs · 2023-01-20

**Decision:**

Reject

**Justification For Why Not Higher Score:**

The main consideration was the issue of novelty and discussion of related work, but the lack of efficiency results was also a significant consideration, as discussed in my review.

**Justification For Why Not Lower Score:**

N/A

**Metareview: Summary, Strengths And Weaknesses:**

The paper proposes a method, ALT, that allows any GNN model to be used as a component within a heterophily-aware approach. The approach combines signals in a learnable way from two complementary filters, in order to obtain an adaptive filter; they further develop a locally adaptive approach utilizing a structure-learning based augmenter.

In general, reviewers find the paper well-written and well-motivated. However, they raise a number of key issues:

- Novelty and discussion of related work: reviewers noted that the novelty is relatively limited considering that there are a number of existing approaches also based on combining graph signals. The related work mentions a number of such methods (e.g. ACM-GNN, FAGCN, GBK), but is quite brief, making it hard to understand the relationships between the current work and these (other than FAGCN in section 3.4). In addition, while the paper notes that only one structure learning solution for heterophily exists (WRGAT), it still leaves the question of how the proposed structure learning approach compares to this and other structure learning approaches.

- Efficiency: the paper does not compare its efficiency with existing approaches; while understandable in that efficiency is not the main contribution and focus of this paper (as the authors state in the rebuttal), I find that since the goal of the paper is to be combined with a broad range of existing GNNs, efficiency still a factor in whether this can practically be done, particularly if there are large differences in efficiency. Particularly, the structure learning / bilevel optimization inclusions are relatively large modifications whose influence on computational cost could (potentially) be larger.

- A few other issues were raised that were satisfactorily addressed by the authors during the rebuttal, particularly the use of validation set data in the bilevel optimization process, and baselines such as ACMII-GCN and LINKX. I thank the authors for their efforts in addressing the concerns and improving the paper.

In the end, reviewers and AC agree that while the work is promising, due to the first two issues, the work is not yet ready for publication at ICLR. The reviews offer a number of helpful suggestions for improvement, so I encourage the authors to continue improving the paper based on the reviews for future submissions.